# Magnon thermal Hall effect via emergent SU(3) flux on the antiferromagnetic skyrmion lattice

Hikaru Takeda [1] ✉, Masataka Kawano[2] ✉, Kyo Tamura[1], Masatoshi Akazawa[1], Jian Yan [1], Takeshi Waki [3], Hiroyuki Nakamura [3], Kazuki Sato[4], Yasuo Narumi [4], Masayuki Hagiwara [4], Minoru Yamashita [1] & Chisa Hotta[5]

Complexity of quantum phases of matter is often understood theoretically by using gauge structures, as is recognized by the $\mathbb{Z}_2$ and U(1) gauge theory description of spin liquids in frustrated magnets. Anomalous Hall effect of conducting electrons can intrinsically arise from a U(1) gauge expressing the spatial modulation of ferromagnetic moments or from an SU(2) gauge representing the spin-orbit coupling effect. Similarly, in insulating ferro and antiferromagnets, the magnon contribution to anomalous transports is explained in terms of U(1) and SU(2) fluxes present in the ordered magnetic structure. Here, we report thermal Hall measurements of $MnSc_2S_4$ in an applied field up to 14 T, for which we consider an emergent higher rank SU(3) flux, controlling the magnon transport. The thermal Hall coefficient takes a substantial value when the material enters a three-sublattice antiferromagnetic skyrmion phase, which is in agreement with the linear spin-wave theory. In our description, magnons are dressed with SU(3) gauge field, which is a mixture of three species of U(1) gauge fields originating from the slowly varying magnetic moments on these sublattices.

Quantum phases of matter are very often complex and require mathematical ingenuity to clarify the nature of emergent phenomena. One famous example is the bound states of kinks that appear as gapped low energy excitations of the seemingly simple Ising spin system in $CoNb_2O_6$, which turned out to follow an emergent $E_8$ exceptional Lie algebra symmetry[1] based on the integrable field theory[2]. Indeed, there are several other cases that effective theories explaining the low-energy excitations of interesting quantum phases are not the simple bosonic or fermionic quasi-particles but are those subject to gauge fields. In the Kitaev model, the spin-1/2 degrees of freedom separate into Majorana fermions and fluxes of an emergent $\mathbb{Z}_2$ gauge field[3]. The half-quantized thermal Hall conductivity reported in $\alpha$-RuCl$_3$ is argued to fit the picture of being carried by these auxiliary fractionalized

Majorana fermions[4–7]. In the pyrochlore systems, quantum fluctuations transform a spin ice state to a U(1) spin liquid phase characterized by the emergent lattice electrodynamics with U(1) global gauge symmetry. The masked pinch point singularities of inelastic neutron scattering experiments on $Pr_2Zr_2O_7$ is considered relevant to this state hosting a monopole excitation[8].

When one refers to the gauge fields in material solids, they are quantum mechanical as there is always a redundancy in the description of phases of wave functions of the related particles. While these gauge fields do not break their symmetries in the way that the lattice symmetries do at the phase transitions, the related *gauge-invariant quantities* can play another important role. For example, the effect of spin-orbit coupling of electrons or non-coplanar structured magnetic

[1]Institute for Solid State Physics, University of Tokyo, Kashiwa 277-8581, Japan. [2]Department of Physics, Technical University of Munich, 85748 Garching, Germany. [3]Department of Materials Science and Engineering, Kyoto University, Kyoto 606-8501, Japan. [4]Center for Advanced High Magnetic Field Science (AHMF), Graduate School of Science, Osaka University, Toyonaka, Osaka 560-0043, Japan. [5]Department of Basic Science, University of Tokyo, Meguro-ku, Tokyo 153-8902, Japan. ✉e-mail: takeda.hikaru@issp.u-tokyo.ac.jp; masataka.kawano@tum.de

moments can be well described as an emergent U(1) gauge field. This gauge field represents the fictitious magnetic field which bends the motion of charges and yields an anomalous Hall effect[9–12]. There, the quantized Hall resistance is explained by the gauge-invariant quantity referred to as a Chern number. In metallic chiral magnets such as MnSi[13,14], the conduction electrons feel an emergent gauge field as they travel through the spatially varying spin texture, which gives a good interpretation of the topological Hall effect[15,16].

Magnons in insulating magnets are dealt as simple bosonic excitations but can also carry a U(1) gauge; the thermal Hall effect in ferromagnetic insulators[17] was the first to report the anomalous transport of magnons due to U(1) gauge field. Theories showed that the antisymmetric Dzyaloshinskii–Moriya (DM) spin exchange interactions or non-coplanar structures of ordered moments can be represented by the gauge field that bears the Berry curvature in the magnon bands[18–21]. Furthermore, in the insulating versions of skyrmions, GaV$_4$Se$_8$[22], the magnon thermal Hall effect is explained by the U(1) gauge field[23] similarly to the cases of metallic skyrmions.

One may thus expect a more abundant gauge structure to appear as useful in the transport phenomena. However, even for simple two-sublattice insulating antiferromagnets in noncentrosymmetric crystals where the U(1) gauge picture is not applicable, it was only recently recognized that there can be another route using the SU(2) gauge field to describe the anomalous thermal Hall effect[24–26].

Here, we report the experimental observation of the thermal Hall effect in the three-sublattice antiferromagnetic skyrmion lattice (AFM-SkL) realized in MnSc$_2$S$_4$. Our large unit cell spin wave theory calculations show that the heat carriers can be described as the magnons in a complex SU(3) gauge field originating from the three sublattice structure.

The AFM-SkL is a new class of skyrmion, recently discovered in a spinel compound MnSc$_2$S$_4$. As shown in Fig. 1a, the Mn$^{2+}$ ($S = 5/2$) ions form a diamond lattice[27,28] and undergo three successive magnetic transitions in a zero field, starting at $T \leq T_N = 2.3$ K from a modulated collinear phase to an incommensurate phase and finally showing a helical magnetic long-range order below 1.6 K[29,30]. Interestingly, the phase above $T_N$ is a correlated paramagnet, which can be described as a classical spiral-spin liquid, having highly degenerate manifold of states with a series of continuous wave numbers forming surface in the reciprocal space[29,31]. When a magnetic field $B$ is applied along the [100], [110], and [111] directions, the helical phase transforms to the triple-$Q$ phase at around $B = 4$–6 T. Combining the neutron scattering experiment and Monte Carlo simulation, the triple-$Q$ state in $B \parallel$ [111] is identified as an AFM-SkL phase, which consists of three sublattices approximately forming a 120°-types of antiferromagnetic order. The structure shown in Fig. 1b is a cross-section of the diamond lattice forming a triangular lattice and stacking in the [111]-direction with six-fold periodicity (64 sites per triangular lattice layer, $N_s = 384$ sites per

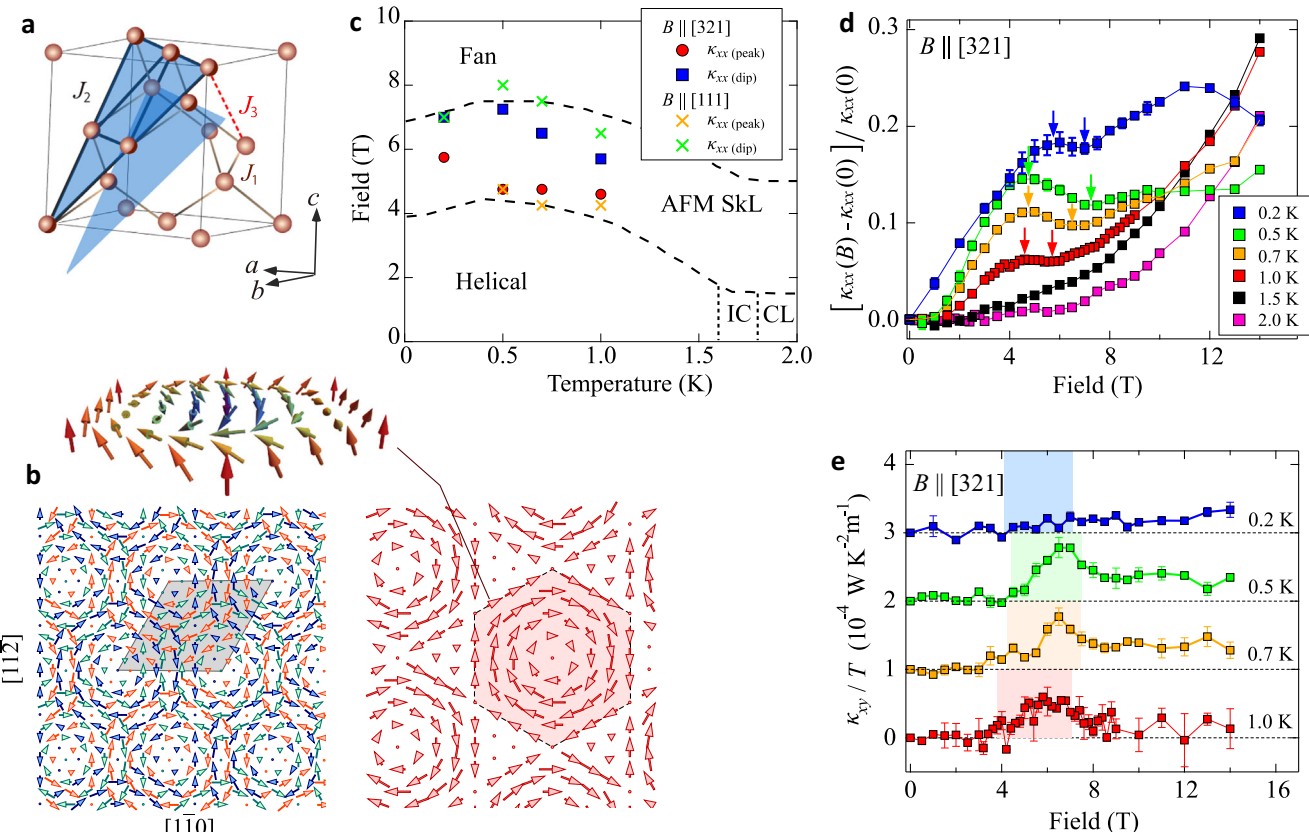

**Fig. 1 | Antiferromagnetic skyrmions in MnSc$_2$S$_4$ and its thermal conduction.**
**a** Diamond lattice formed by Mn$^{2+}$ ions (red circles) in MnSc$_2$S$_4$. The two [111]-planes marked in blue are parallel to each other and include Mn$^{2+}$ ions belonging to different sublattice of the bipartite diamond structure. Inside the plane these ions form a triangular lattice. The magnetic unit cell of AFM-SkL consists of six triangular layers. **b** Schematic figure of the AFM-SkL state in MnSc$_2$S$_4$ viewed along the [111] direction (left-bottom panel), where the spins on three different sublattices are shown in different colors, and the unit cell on that layer including 64 sites are shown. Spins on one of the sublattices forming a ferromagnetic skyrmion lattice are extracted on the right panel, and its hexagonal unit is shown in more detail on the

top panel. **c** Magnetic field $B$ versus temperature $T$ phase diagram of MnSc$_2$S$_4$. Broken lines are phase boundaries determined by the neutron diffraction measurements[30]. Filled circles and squares are the peaks and dips of $\kappa_{xx}$ for field direction $B \parallel$ [321], while crosses are those for $B \parallel$ [111]. **d** Field dependence of thermal conductivity $\kappa_{xx}$ normalized by its value at $B = 0$ T at $T \leq 2$ K. The arrows indicate the positions of peaks and dips, plotted in panel (**c**). **e** Field dependence of $\kappa_{xy}/T$ in the magnetically ordered phases. The shaded area represents the AFM-SkL phase[29,30]. All the data points in the normalized $\kappa_{xx}$ and $\kappa_{xy}/T$ are the ones averaged over the field-up and field-down measurements, and the error bars are the maximum deviations of the data from the averages.

unit cell). Each sublattice forms a triangular SkL, which can be well-explained theoretically within the classical framework[32–34].

# Results

## Phase diagram

We performed thermal transport measurements on the single crystals of $MnSc_2S_4$ using the setup shown in Supplementary Note 1A. The detailed analysis of the field, temperature, and sample dependences are presented in Supplementary Note 1B–E. Figure 1c shows the $B-T$ phase diagram of $MnSc_2S_4$ at low temperature. The data points indicate the location of peaks and dips in the field dependence of the thermal conductivity ($\kappa_{xx}$) for two different field directions, $\boldsymbol{B} \parallel$ [321] and $\boldsymbol{B} \parallel$ [111], shown in Fig. 1d. They are not much sensitive to the field direction, and show good agreement with the phase boundaries of the AFM-SkL phase obtained previously by the neutron diffraction experiment for $\boldsymbol{B} \parallel$ [111][30]. Although the temperature dependence of $\kappa_{xx}$ does not show a clear anomaly at $T_N$ (see Supplementary Fig. 3), the field dependence has distinct upturn and downturn, which are more visible for lower temperatures and disappear at $T \gtrsim T_N$. The good correspondences of these features with the phase boundaries indicate that $\kappa_{xx}$ is strongly influenced by the magnetic ordering.

## Thermal Hall measurement

Figure 1e shows the field dependence of thermal Hall conductivity $\kappa_{xy}/T$ at $T \leq 1$ K for $\boldsymbol{B} \parallel$ [321] of $MnSc_2S_4$. In the helical phase at $B \lesssim 4$ T, $\kappa_{xy}/T$ is suppressed to nearly zero or slightly negative values, which, however, shows an abrupt and substantial increase in entering the AFM-SkL phase at around 4 T. Its amplitude is overall suppressed at the lowest temperature 0.2 K, which is usual for the thermal Hall conductivity that relies on thermally driven bosonic excitations. At above ~8 T, where the previous theory predicts a fan phase, $\kappa_{xy}/T$ becomes suppressed but remains positive and finite.

## Linear spin wave theory with a large unit cell

Unlike the standard insulating ferro or antiferromagnets, whether and when the thermal Hall conductivity becomes finite in SkL is not well understood. The spin-wave calculation for Néel type (ferro)-SkL on a triangular lattice is performed for a model with Heisenberg and DM interactions and single-ion uniaxial anisotropy[35–37], while they did not consider the transport properties. The thermal Hall effect observed in the ferromagnetic Néel SkL in $GaV_4Se_8$ is studied by the phenomenological U(1) gauge theory, showing a good agreement with the experimental data[22]. However, their Chern number and the Berry curvature contradict those of the spin-wave theory. Indeed, the Berry curvature depends much on the details of the Hamiltonian and the inter-band transition, and accordingly, the same SkL structure does not necessarily yield the same Berry curvature. In such a case, the simplest U(1) gauge theory may not be sufficient.

For these reasons, we seriously perform a spin-wave theory by taking advantage that the microscopic lattice model that reproduces well the experimental observations in $MnSc_2S_4$ are derived based on the Monte Carlo simulation[30], which is given as

$$\mathcal{H} = \sum_{\boldsymbol{r},\boldsymbol{\delta}_l} \frac{J_l}{2}\boldsymbol{S}_{\boldsymbol{r}} \cdot \boldsymbol{S}_{\boldsymbol{r}+\boldsymbol{\delta}_l} + \frac{3}{2}J_{\parallel}\sum_{\boldsymbol{r},\boldsymbol{\delta}_1}(\boldsymbol{S}_{\boldsymbol{r}} \cdot \hat{\boldsymbol{\delta}}_1)(\boldsymbol{S}_{\boldsymbol{r}+\boldsymbol{\delta}_1} \cdot \hat{\boldsymbol{\delta}}_1) \\ + A_4\sum_{\boldsymbol{r},\mu=x,y,z}(S_{\boldsymbol{r}}^{\mu})^4 - g\mu_B\sum_{\boldsymbol{r}}\boldsymbol{B} \cdot \boldsymbol{S}_{\boldsymbol{r}}. \tag{1}$$

Here, $S = 5/2$, $(J_1, J_2, J_3) = (-0.31, 0.46, 0.087)$ K are the Heisenberg exchange interactions, and $\boldsymbol{\delta}_l$ ($l = 1, 2, 3$) are the corresponding vectors representing the first, second, and third neighbors in the diamond lattice ($\hat{\boldsymbol{\delta}}_l$ is the unit vector). The anisotropic coupling constants are set to $J_{\parallel} = 0.01$ K and $A_4 = 0.0016$ K (in unit of temperature)[30]. The magnon dispersions in an applied field are shown in Fig. 2a, b for the helical and fan phases and in Fig. 2c, d for AFM-SkL phases for

two different field directions. The $n$th magnon bands are colored by the Berry curvature $\Omega_{xy}^{(n)}$ they carry. The thermal Hall conductivity is evaluated by integrating $\Omega_{xy}^{(n)}$ as[38],

$$\kappa_{xy} = -\frac{k_B^2 T}{\hbar}\int_{BZ}\frac{d^3\boldsymbol{k}}{(2\pi)^3}\sum_{n=1}^{N_s}c_2[f(\varepsilon_n(\boldsymbol{k}))]\,\Omega_{xy}^{(n)}(\boldsymbol{k}), \tag{2}$$

where $f(\varepsilon) = 1/\{\exp(\varepsilon/k_BT) - 1\}$ is the Bose distribution function, $c_2[x] = \int_0^x dt[\{(1+t)/t\}]^2$, and the integration is carried out over $N_s$-bands in the first Brillouin zone (see Fig. 2).

The calculated field-dependence of $\kappa_{xy}$ is shown in Fig. 2e. It remains small in the helical phase, reflecting the observation in Fig. 2a that $\Omega_{xy}^{(n)}$ has finite contributions on part of the magnon bands, while they are both positive and negative on nearby branches which mostly cancel out. If we perform the calculation by setting $J_{\parallel} = 0$, this small contribution disappears and $\kappa_{xy} = 0$ is obtained (see Supplementary Fig. 7a). This is because such an anisotropic bond-dependent exchange interaction can be the source of the Berry curvature[39].

Contrastingly, for AFM-SkL phase, $\Omega_{xy}^{(n)}$ takes overall large values throughout the whole magnon bands, particularly, a large negative value on the lowest branch when $\boldsymbol{B} \parallel$ [111]. This explains why $\kappa_{xy}$ abruptly increases in entering the AFM-SkL phase. In further increasing the field, the calculated $\kappa_{xy}$ has peaks at 5 and 7 T for $\boldsymbol{B} \parallel$ [111] and at around 5.5 T for $\boldsymbol{B} \parallel$ [321]. The field $\boldsymbol{B} \parallel$ [111] is perpendicular to the SkL plane and has a larger effect than $\boldsymbol{B} \parallel$ [321]; the magnon bands at $\varepsilon \gtrsim 0.5$ K are much dense and $\kappa_{xy}$ are larger.

In the fan phase above 7 T, $\kappa_{xy}$ almost disappears, since $\Omega_{xy}^{(n)}$ (see Fig. 2b) is mostly small except near the zone boundary. This does not explain a finite, almost field-independent $\kappa_{xy}$ in the experiment. It should be noted that the energy scale of $B \gtrsim 8$ T is comparable to or higher than the temperature at which the correlated paramagnetic phase appears at zero field characterized by the coexistent magnetic correlations of several different periods[29]. This competition may transform the system into slightly different types of orderings or another liquid state. Namely, the state may not be the fan phase, which is beyond the description of Eq. (1). Apart from this issue, the consistency of theory and experiment at $B \lesssim 8$ T is sufficient to confirm that AFM-SkL phase yields substantial and stable $\kappa_{xy} > 0$.

## Examination of the thermal conductivity

In magnetic insulators, the carriers contributing to thermal transport can be phonons and magnons. Here, we clarify experimentally that $\kappa_{xx}$ has indeed a substantial contribution from magnons ($\kappa_{xx}^{mag}$) on top of phonons ($\kappa_{xx}^{ph}$). Figure 3a shows the field dependence of $\kappa_{xx}$ at $T > T_N$ to be compared with Fig. 1d at $T \leq T_N$. The positive magnetothermal conductivity observed at lower fields for $T < T_N$ becomes negative above $T_N$.

The major effect of the magnetic field on the present system is to vary the magnon gap. In our theoretical calculation, when the system remains within the same phase the overall shape of the magnon bands does not change much while both the bandwidth and the gap vary (see Supplementary Fig. 7). As shown in Fig. 2e, the magnon gap first decreases toward zero in approaching the helical-to-AFM-SKL phase transition point slightly below 4 T, then increases again on entering the AFM-SkL phase, and closes at another transition point near 7 T. In general, an decrease of magnon gap increases $\kappa_{xx}^{mag}$ since the number of excited magnons that depends on the Bose distribution function increases. This can in turn suppress $\kappa_{xx}^{ph}$, supposing that there is a sufficiently large amount of scattering of phonons by the magnons.

For this reason, the positive magnetothermal conductivity observed at $T < T_N$ in decreasing the gap inside the helical phase is the feature attributed to an increase of $\kappa_{xx}^{mag}$. In the AFM-SkL phase, $\kappa_{xx}$ shows a slight decrease, which should be because of the re-opening of the magnon gap, in addition to a decrease of $\kappa_{xx}^{ph}$ by the extra scattering

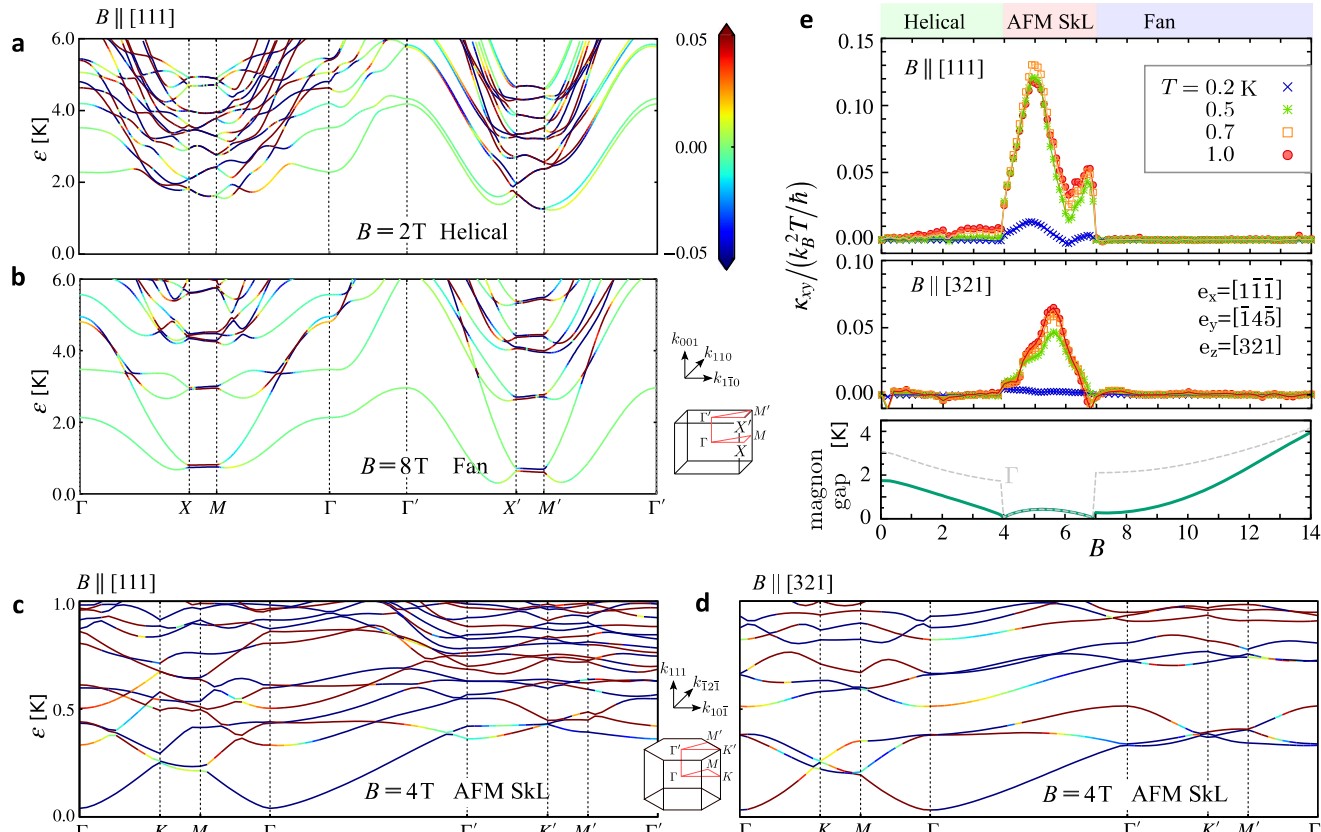

**Fig. 2 | Theoretical calculations of the magnon bands and the thermal Hall conductivity. a–d** Magnon bands obtained by the spin-wave theory for the helical phase at $B = 2$ T ($N_c = 16$ site unit cell, $\boldsymbol{B} \parallel [111]$), the fan phase at $B = 8$ T ($N_c = 16$, $\boldsymbol{B} \parallel [111]$), and the AFM-SkL phase at $B = 4$ T ($N_c = 384$, $\boldsymbol{B} \parallel [111]$ and [321]). Reciprocal space is shown in the inset. Magnon bands are shown with the color density plot of

the Berry curvature $\Omega_{xy}^{(n)}$ at each $\boldsymbol{k}$-point. **e** Field dependent $\kappa_{xy}/T$ for $T = 0.2, 0.5$, and $0.7, 1$ K from the linear-spin-wave theory using the same condition as panels (**a**–**d**). Field direction is taken as $\boldsymbol{B} \parallel [111]$ and [321]. The bottom panel shows the magnon gap (solid line) and the gap at Γ-point (broken line).

effect of phonons by magnetic skyrmions as suggested in the case of ferro-SkL in GaV$_4$Se$_8$[22].

In our experiment at $T = 3$–8 K, there is a negative magnetothermal conductivity observed up to 8 T (Fig. 3a), which should be of magnon origin. It is known that a negative magnetothermal conductivity of $\kappa_{xx}^{\text{ph}}$ is caused by a resonance scattering of phonons with spins[40]. In that case, its field dependence scales with $B/T$, taking the minimum value when the Zeeman energy matches the thermal energy scale, $B \sim 4k_BT$, at which the phonon distribution reaches the maximum. However, the field dependence of $\kappa_{xx}$ clearly does not scale with $B/T$ (Supplementary Fig. 4). Therefore, we speculate the origin of the negative magnetothermal conductivity to be the magnetic excitation from a experimentally reported correlated paramagnet at finite temperature consisting of a manifold of states showing specific diffuse signature in momentum space[29], which is beyond the present theoretical treatment. We further mention that magnons are good quasiparticles known to have a long lifetime at low temperature[41–43], and they suffer resonance scattering with phonons only at high energies where the magnon branches cross the phonon ones. The positive magnetothermal conductivity at high fields is due to the enhancement of $\kappa_{xx}^{\text{ph}}$ caused by the suppression of magnetic fluctuations by the magnetic field.

We finally note that phonons are unlikely the origin of the thermal Hall effect, although $\kappa_{xx}^{\text{ph}}$ is substantial in the AFM-SkL phase. For a thermal Hall effect of phonons, the temperature dependence of $\kappa_{xy}$ is known to scale with that of $\kappa_{xx}$ as observed in several materials[44–46]. It is clearly not the case for MnSc$_2$S$_4$ as shown in Fig. 3b; whereas $\kappa_{xx}/T$ monotonically decreases as lowering $T$, $\kappa_{xy}/T$ shows a peak at around

$T_N/2$. This supports the magnon origin for the thermal Hall effect in the AFM-SkL phase.

## Discussion

We first discuss the correspondence of the experiment in Fig. 1e and the theory in Fig. 2e for $\boldsymbol{B} \parallel [321]$. The common features are captured as follows; at 0.2 K, $\kappa_{xy}$ remains small and structureless, while for 0.5, 0.7, and 1 K, there is a single peak at around 5.5–6 T and the amplitudes no longer differ much for different temperatures. To understand these features, we want to know up to how many levels counted from the bottom the magnons are excited to give major contributions to $\kappa_{xy}$ (Supplementary Fig. 8). At 0.2 K, from among totally 384 magnon bands, the lowest 24 levels corresponding to the energy window of up to 2 K are the major excitations for $\kappa_{xy}$, while at 0.5 K, the excitation goes much higher. This means that the dense magnon band structure of skyrmions up to very high energies plays a crucial role in the magnon transport. Accordingly, although the magnon gap increases toward 5 T and then decreases with a field, $\kappa_{xy}$ is not sensitive to the degree of magnon gap. The nontrivial distribution of $\Omega_{xy}^{(n)}$ up to the high energy determines the field dependence of $\kappa_{xy}$.

Indeed, the importance of dealing with precise spin-wave magnon bands derived from the individual spin Hamiltonian is clear even for the simple ferromagnetic Néel SkL phase of GaV$_4$Se$_8$[22]. Although the phenomenological U(1) gauge theory gives $\kappa_{xy}$ in good agreement with the experimental data, the corresponding Chern number contradicts those obtained by the spin-wave theory[35,36] at the lowest two magnon bands. The inconsistency naturally arises since phenomenological U(1) gauge theory employs the oversimplied hopping Hamiltonian of

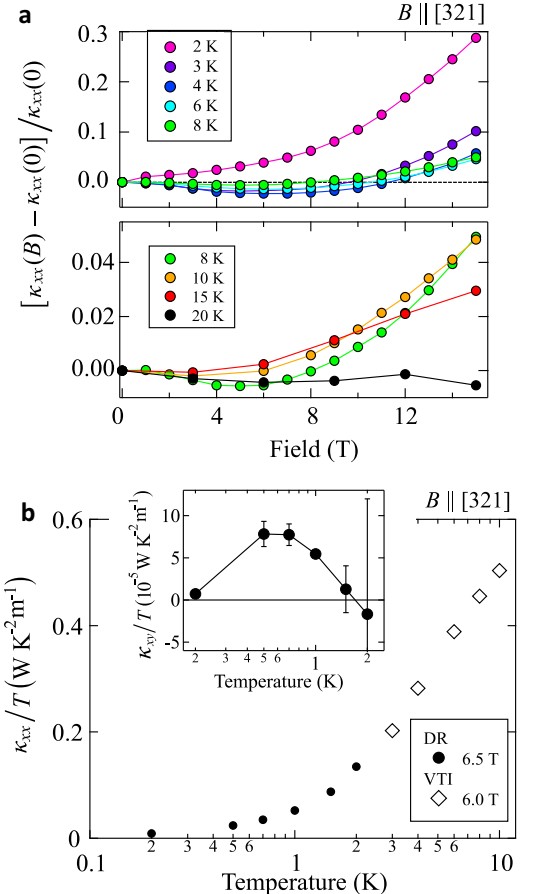

**Fig. 3 | Field and temperature dependence of the longitudinal thermal conductivity. a** Field dependence of $\kappa_{xx}(B)$ normalized by the zero-field value above $T_N$. **b** Temperature dependence of $\kappa_{xx}/T$ and $\kappa_{xy}/T$ (inset) in the AFM-SkL phase (6.0 and 6.5 T) obtained by the dilution refrigerator (DR) ($T < 3$ K) and the variable temperature insert (VTI) ($T > 3$ K) measurements. The error bars for $\kappa_{xy}/T$ are maximum deviations of the data from the averaged data on the field-up and field-down measurements.

bosons with the U(1) Peierls phase, which does not reflect many details of the original spin Hamiltonian, e.g., dropping off the particle-non-conserving terms. Let us illustrate this point through the derivation of the U(1) gauge in the ferro-SkL[23,47]; the slowly varying magnetic moment is expressed by a single field operator $\hat{s}(r)$, and the standard Heisenberg Hamiltonian on a triangular lattice with lattice spacing $a$ is taken the continuum limit by $\hat{S}_i \rightarrow \nu \hat{s}(r)$ with unit cell volume $\nu = \sqrt{3}a^2/2$;

$$\mathcal{H}_{\text{eff}}^{\text{FM}} \sim 3J\nu \int d^2r \, \hat{s}^t(r)\left(1 + \frac{a^2}{4}\nabla^2\right)\hat{s}(r). \tag{3}$$

The bosons are introduced by the Holstein–Primakoff transformation as

$$\hat{s}(r) \sim \sqrt{\frac{S}{2\nu}}\left((b_r + b_r^\dagger)e^x(r) - i(b_r - b_r^\dagger)e^y(r)\right) + \left(\frac{S}{\nu} - b_r^\dagger b_r\right)e^z(r), \tag{4}$$

where the unit vectors $e^\mu(r)(\mu = x, y, z)$ form a *local* orthogonal coordinate, with $e^z(r) \equiv m(r)$ pointing in the direction of the ordered local moment specified by the angles $(\theta(r), \phi(r))$ (see Fig. 4a). Substituting Eq. (4) to Eq. (3) and taking the lowest-order derivative of $e^\mu$, which

drops off the particle-nonconserving terms like $b_r b_r$, we obtain

$$\mathcal{H}_{\text{eff}}^{\text{FM}} \sim 6J\nu \int d^2r \, b_r^\dagger(\nabla - iA(r))^2 b_r, \tag{5}$$

where a fictitious U(1) vector potential $A(r) = -\cos\theta(r)\nabla\phi(r)$ is generated by the variation of angles $\theta$ and $\phi$ (see Fig. 4a). When we put it back to the bosons $b_i$ on lattice sites, we obtain $\mathcal{H}_{\text{eff}}^{\text{FM}} \sim JS\sum_{i,j} U_{ij} b_i^\dagger b_j$ with the U(1) gauge field given as $U_{ij} = \exp(i\int_{r_i}^{r_j} dr \cdot A(r))$.

For the intuitive understanding of the origin of thermal Hall effect, the underlying gauge structures play a crucial role as summarized in Fig. 4b. For ferromagnets, *a gauge-invariant quantity* is a flux $\phi$ that is generated from the U(1) gauge field when the magnons hop around the closed loop as, $U_{12}U_{23}U_{34}U_{41} = e^{i\phi}$. This flux works as internal field and bends the motion of magnons, yielding $\kappa_{xy} \neq 0$. However, in the square or triangular lattices, the adjacent loop, $(U_{12}U_{23'}U_{3'4'}U_{4'1})^* = e^{-i\phi}$, generates the same amount of flux with opposite sign, and since this flux pattern is invariant under the symmetry operation of time reversal $\phi \rightarrow -\phi$ combined with the translation by one lattice spacings, the contributions of the fluxes cancel out and yield $\kappa_{xy} = 0$[19]. Because of this no-go theorem for edge shared lattices, the thermal Hall effect was allowed only in pyrochlore and other corner shared lattices[19].

However for antiferromagnets, the SU(2) flux can be generated as another *gauge-invariant quantity*. On a square lattice (Fig. 4c), the modulation of spins on the two magnetic sublattices are independently represented by the field operators and the corresponding two species of magnons are introduced, which are regarded as those of up and down pseudo-spins. Similarly to ferromagnets, even when each magnon feels the fictitious U(1) gauge field created on the corresponding sublattice, its effect cancels out. Whereas if there exists an anti-symmetric exchange coupling or the canting of moments due to magnetic fields they work as "pseudo-spin-orbit coupling" between the two species of magnons, represented in the form of SU(2) gauge field[24–26]. In analogy with the SU(2) hopping of Rashba electrons of semiconductors[48], this gauge field allows a nonzero thermal Hall effect on a square lattice antiferromagnet[25]. Let us choose a translation-invariant gauge field $T_{ij} = e^{i\Theta_\alpha}$ as a natural representation with $2 \times 2$ matrix $\Theta_x$ and $\Theta_y$ for the hopping in the $+x$- and $+y$-bond directions, respectively. Due to the non-commutativity of the SU(2) gauge field, we find $T_{12}T_{23}T_{34}T_{41} \equiv e^{i\Phi \cdot \sigma} = e^{i\Theta_x}e^{i\Theta_y}e^{-i\Theta_x}e^{-i\Theta_y} \sim e^{-[\Theta_x, \Theta_y]}$ with Pauli matrices $\sigma$ at the lowest order which gives the gauge invariant SU(2) flux $|\Phi|$. This time, the flux in the adjacent plaquette, $T_{12}T_{23'}T_{3'4'}T_{4'1} \equiv e^{-i\Psi \cdot \sigma} \sim e^{+[\Theta_x, \Theta_y]}$, gives $\Psi \simeq \Phi$. This means that the net flux is no longer zero and cannot be eliminated by any symmetry operation, namely the time reversal symmetry is broken, which is the reason for having $\kappa_{xy} \neq 0$. The SU(2) magnon thermal Hall effect is not established yet in experiment, however, a thermal Hall effect observed in an antiferromagnetic phase of the Kitaev candidate material, $Na_2Co_2TeO_6$, might be related, which needs to be scrutinized in further studies[49].

For the three-sublattice antiferromagnets, there exist at least three field operators $\hat{s}_\ell(r)(\ell = A, B, C)$ that describe the spatial variation of moments (Fig. 4e), and the corresponding magnons form three-component pseudo-spins. In the present AFM-SkL, these sublattices equivalently generate U(1) gauge fields as $A_\ell(r) = -\cos\theta_\ell(r)\partial_\mu\phi_\ell(r)$, which we actually calculated as shown in Fig. 4e. The fictitious U(1) fields shown as density plots are given as $b_\ell^z(r) = \partial_x A_\ell^y(r) - \partial_y A_\ell^x(r)$ (see Supplementary Note 2D for details of calculation). The locations where $m(r)$ points in the $z$-direction ($\theta = 0, \pi$) serve as vortex centers of $A_\ell(r)$ and generate large $b_\ell^z(r)$. Now if the three U(1) gauge fields are decoupled, their effect cancel out on the whole. However, there naturally arises a coupling that

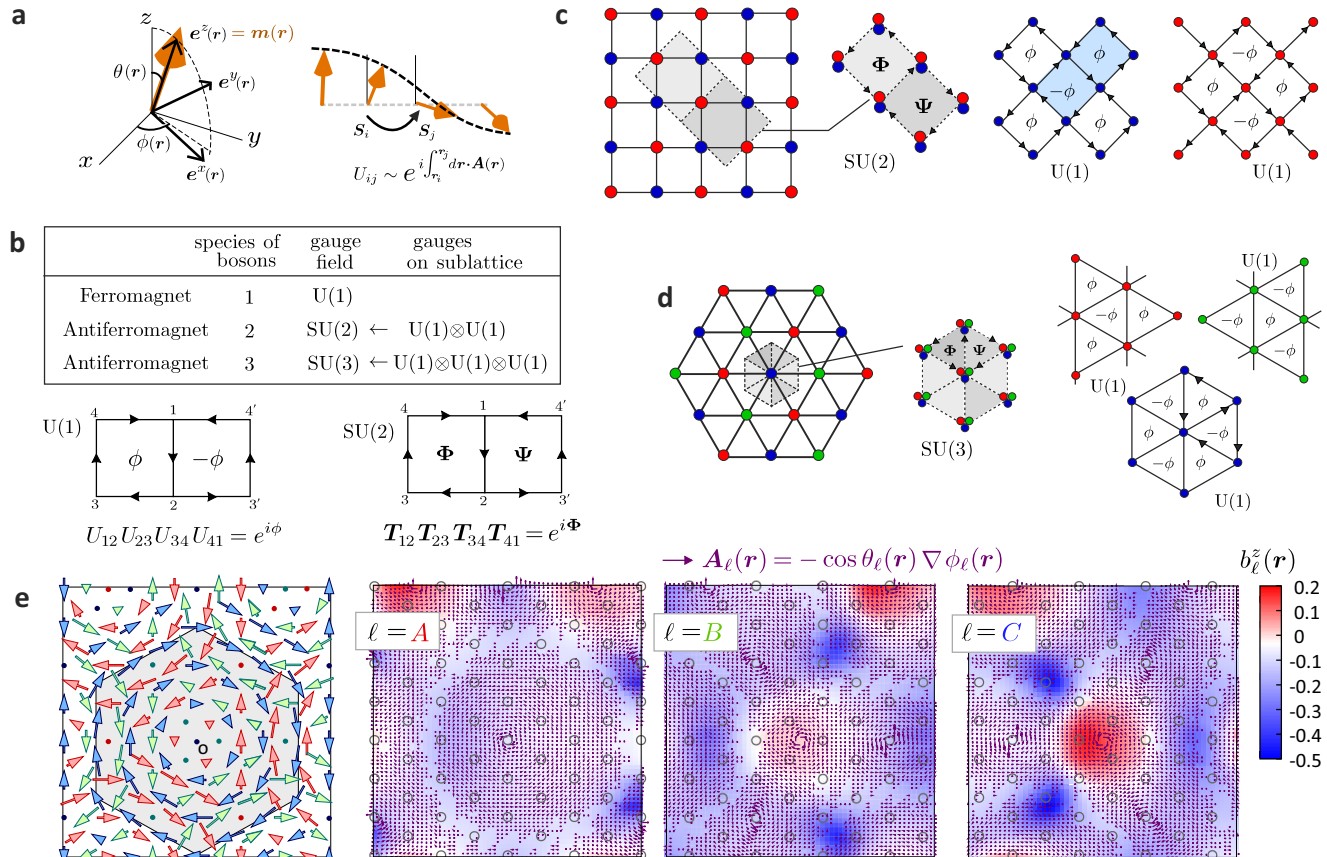

**Fig. 4 | Illustration of how the gauge structures are linked to magnons.**
**a** Schematic description of the ordered spin moments slowly varying in space. The $x, y, z$-components of spin $\boldsymbol{S}(\boldsymbol{r})$ point toward $(\boldsymbol{e}^x(\boldsymbol{r}), \boldsymbol{e}^y(\boldsymbol{r}), \boldsymbol{e}^z(\boldsymbol{r}))$ which form the local coordinate with $\boldsymbol{e}^z(\boldsymbol{r}) = \boldsymbol{m}(\boldsymbol{r})$ specified by the angle $(\theta(\boldsymbol{r}), \phi(\boldsymbol{r}))$. Gradient of $\phi(\boldsymbol{r})$ generates a U(1) gauge field. **b** Relationships of underlying magnetic orderings, the number of species of magnons that naturally describe the low energy excitations, and the gauge structures that couple to magnons. The lower panels give the description of the U(1) flux and SU(2) flux for two adjacent plaquettes on the square lattice. The U(1) gauge field, $U_{ij} = e^{i\theta_{ij}}$, with a Peierls phase, and the SU(2) gauge field, $\boldsymbol{T}_{ij} = e^{i\boldsymbol{\Theta}_{ij}}$ in the form of $2 \times 2$ matrix that mixes the two components on hopping along $i \to j$ bond. When magnons hop around the plaquette along the arrows, they acquire a nonzero phase as gauge-invariant flux $\phi$. For U(1) gauge field $\phi$ and $-\phi$ on

adjacent plaquettes cancel out due to symmetry and give zero thermal Hall effect, whereas the SU(2) gauge field leads to a nearly uniform flux $\boldsymbol{\Phi}$ due to non-commutativity and give finite thermal Hall effect. **c** Square lattice antiferromagnet with two magnetic sublattices each generating the U(1) gauge field. When they couple and form an SU(2) gauge field, we find nonzero thermal Hall effect. **d** Three sublattice triangular lattice antiferromagnets and their three U(1) gauge fields which can couple and form an SU(3) gauge field. **e** Vector potentials $\boldsymbol{A}_\ell(\boldsymbol{r})$ representing the U(1) gauge field of $l = A, B, C$ sublattice calculated for our AFM-SkL structure, and the corresponding field $b^z_l(\boldsymbol{r})$. Left panel is the corresponding AFM-SkL structure on the triangular lattice colored in red, green and blue for $\ell = A, B, C$ sublattices, respectively.

converts them to the SU(3) gauge field. We start from the simplest Heisenberg Hamiltonian by discarding the magnetic anisotropy terms of Eq. (1), which does not deteriorate the context;

$$\mathcal{H}_{\text{eff}}^{\text{AFM}} \sim \frac{J\nu}{2} \int d^2\boldsymbol{r} \sum_{\ell,\ell'} \hat{s}^t_\ell(\boldsymbol{r})\left(1 + \frac{a^2}{4}\nabla^2\right)\hat{\boldsymbol{s}}_\ell(\boldsymbol{r}). \tag{6}$$

Using the three Holstein–Primakoff bosons $b_\ell(\boldsymbol{r})$ with indices $\ell = A, B, C$ added to Eq. (4), we reach the following effective Hamiltonian for the AFM-SkL,

$$\mathcal{H}_{\text{eff}}^{\text{AFM}} \sim \int d^2\boldsymbol{r}\, \Phi^\dagger(\boldsymbol{r})\left[\boldsymbol{F}(\boldsymbol{r})a^2\nabla^2 + \sum_{\mu=x,y}\boldsymbol{G}_\mu(\boldsymbol{r})a\partial_\mu\right]\Phi(\boldsymbol{r}), \tag{7}$$

where $\Phi^\dagger(\boldsymbol{r}) = (b^\dagger_A, b_A, b^\dagger_B, b_B, b^\dagger_C, b_C)$ and $\boldsymbol{F}(\boldsymbol{r})$ and $\boldsymbol{G}_\mu(\boldsymbol{r})$ are $6 \times 6$ matrices consisting of $\boldsymbol{e}^\mu_\ell \cdot \boldsymbol{e}^\mu_{\ell'}$ and $\boldsymbol{e}^\mu_\ell \cdot \partial_\mu \boldsymbol{e}^\mu_{\ell'}$, respectively, with $\ell \neq \ell'$. The diagonal elements of $\boldsymbol{G}_\mu(\boldsymbol{r})$ are proportional to the U(1) gauge fields $\boldsymbol{A}_\ell(\boldsymbol{r})$ in Fig. 4e. Despite the simple form of Eq. (6), a substantial off-diagonal elements in $\boldsymbol{G}_\mu(\boldsymbol{r})$ exist and transform the three U(1)'s to the SU(3) gauge field. Let $\Psi^\dagger(\boldsymbol{r})$ be the operators

obtained by the unitary transformation to $\Phi^\dagger(\boldsymbol{r})$ that diagonalizes $\boldsymbol{F}(\boldsymbol{r})$, where we find the form

$$\mathcal{H}_{\text{eff}}^{\text{AFM}} \sim \int d^2\boldsymbol{r}\, \Psi^\dagger(\boldsymbol{r})\left[\sum_{\mu=x,y}\left(I_{3\times3}\partial_\mu + \boldsymbol{T}_\mu(\boldsymbol{r})\right)^2 \otimes \tau^x\right]\Psi(\boldsymbol{r}), \tag{8}$$

where $\boldsymbol{G}_\mu(\boldsymbol{r})$ is converted to the $3 \times 3$ matrix $\boldsymbol{T}_\mu(\boldsymbol{r})$ that serves as a high-rank vector potential or the SU(3) gauge field, as we find from the analogy with Eq. (5).

We point out that the in-plane uniform 120° magnetic ordering no longer creates a U(1) gauge field, neither an SU(3) gauge field, and has $\kappa_{xy} = 0$. Therefore, although the SU(3) gauge field itself remains a fictitious object, its non-commutativity forces the gauge-invariant SU(3) flux to break the time reversal symmetry combined with lattice symmetry and thus plays a key role to understand why the AFM-SkL has a nonzero thermal Hall effect.

The SU(3) gauge has been a theoretical object for describing the interactions of quarks or gluons in particle physics, and not much has been discussed in condensed matter, particularly in experiment. While the SU(3) symmetry can sometimes appear, e.g., in cold atoms[50] and

the SU(3) Heisenberg Hamiltonian may yield exotic phases[51], the present system would be the first to observe experimentally the phenomena that directly reflect the SU(3) gauge structure in material solids.

## Methods

### Experimental

Single crystals of $MnSc_2S_4$ were synthesized by the chemical vapor transport method. We call two crystals with the shape of a thin plate with [321] plane as sample #1 and [111] plane as sample #2. The thermal-transport measurements were performed by the steady method by using a variable temperature insert (VTI) for 2–60 K and a dilution refrigerator (DR) for 0.1–3 K. The heat current $J_Q$ was applied along [$\bar{1}11$] ([$2\bar{1}\bar{1}$]) and the magnetic field $\boldsymbol{B}$ was applied along [321] ([111]) for the sample #1 (#2). The detailed setup of the thermal transport measurements is shown in Supplementary Note 1A, where we show that the temperature slope within the sample is stable within few percent of the sample temperature. The errors of $\kappa_{xx}$ and $\kappa_{xy}$ are evaluated from systematic errors between measurements conducted under almost the same condition (field-up and field-down measurements at the same temperature). In the main text, we show the experimental data for the sample #1. The amplitude of $\Delta\kappa_{xx}(B) = \kappa_{xx}(B) - \kappa_{xx}(0)$ and $\kappa_{xy}$ differ between sample #1 and #2 consistently by a factor of 9–10, which is safely attributed to the nonmagnetic contributions from phonons due to sample quality. For the field dependence, we find that both samples show essentially the identical results by incluing this factor (see Supplementary Note 1D), indicating high reproducibility of our results, and also confirming that the two field directions yield the same magnetic phases.

### Calculation

We performed a spin-wave theory for Eq. (1) for parameters where the magnetic orderings of large spatial periods take place. We adopted the magnetic structures of the helical and fan phases as $\boldsymbol{m}_j \propto -\sin(\boldsymbol{q} \cdot \boldsymbol{r}_j)\boldsymbol{e}_{1\bar{1}0} - \cos(\boldsymbol{q} \cdot \boldsymbol{r}_j + \phi)\boldsymbol{e}_{110} + M\boldsymbol{e}_B$ with $\boldsymbol{q} = 3\pi/2(1, 1, 0)$ and $\phi = -\pi(\text{fan})$ and $-3\pi/2(\text{helical})$. For AFM-SkL, $\boldsymbol{m}_j \propto \sum_l \sin(\boldsymbol{q}_l \cdot \boldsymbol{r}_j)\boldsymbol{e}_l - \cos(\boldsymbol{q}_l \cdot \boldsymbol{r}_j - 9\pi/8)\boldsymbol{e}_{111} + M\boldsymbol{e}_B$ with $\boldsymbol{q}_1 = 3\pi/2(1, -1, 0)$, $\boldsymbol{q}_2 = 3\pi/2(1, 0, -1)$, $\boldsymbol{q}_3 = 3\pi/2(0, 1, -1)$, for $\boldsymbol{e}_l = (\bar{1}, \bar{1}, 2), (1, \bar{2}, 1)$ and $(2, \bar{1}, \bar{1})$. The amplitude of $M$ is determined to minimize the summation of classical ($E_{cl}$) and quantum corrections ($E_{qc}$). We perform a local gauge transformation to rotate the local spins to $z$-direction, and apply a Holstein–Primakoff transformation[52] in the rotating frame, solving the resultant spin-wave Hamiltonian represented by the $2N_s \times 2N_s$ matrix, where we take $N_s = 16$ for the helical phase and $N_s = 384$ for the AFM-SkL phase[53]. For more details of the results, see Supplementary Note 2.

## Data availability

The data that support the findings of this study are included in this published article and Supplementary Information files. Source data are provided as a Source Data file. Source data are provided with this paper.

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

## Acknowledgements

The work is supported by JSPS KAKENHI Grants Grants No. JP19H01848 (M.Y.), No. JP23H01116 (M.Y.), No. JP21H05191 (C.H.), No. JP21K03440 (C.H.), No. JP21H01035 (Y.N.), and No. JP17H06137 (Y.N.) from the Ministry of Education, Science, Sports and Culture of Japan and the Murata Science Foundation. M.K. was supported by JSPS Overseas Research Fellowship.

## Author contributions

H.T., K.T., M.A., J.Y. and M.Y. performed the thermal-transport measurements and analyzed the data. The samples were synthesized and characterized by K.S., Y.N., M.H., T.W. and H.N. M.K. performed the spin-wave calculation and constructed the effective theory with C.H. C.H. wrote the manuscript with the input from all authors. H.T. and M.K. contributed equally to the work.

## Competing interests

The authors declare no competing interests.
