## [Peer Review File · Nature Communications]

Magnon thermal Hall effect via emergent SU(3) flux on the antiferromagnetic skyrmion latticeREVIEWER COMMENTS

Reviewer #1 (Remarks to the Author):

The authors have systematically studied the spin textures using large-scale spin wave theory and the magnon transport through thermal Hall effect measurements. The authors identified a regime that hosts antiferromagnetic skyrmion and unconventional thermal Hall effect and then attributed it to the emergence of SU(3) gauge symmetry. The authors have discussed the contribution from phonon and concluded that its contribution to the unconventional thermal Hall effect is minimal. The referee finds that it is interesting and significant in general. To meet the high standard of the Nature brand journal, the referee has the following comments and suggestions.

1. The authors point out the importance of symmetry, U(1), SU(2), and SU(3), in describing the excitation of solid-state systems. However, there is no schematic illustration for these symmetries and how they are linked to the magnon in this work. The referee suggests that the authors need to draw illustrations to highlight this link.
2. The authors point out that the antiferromagnetic skyrmion phase supports SU(3) symmetry, which is in contrast to U(1) symmetry in the ferromagnetic skyrmion phase. In general, antiferromagnetic skyrmion should have degenerate magnon bands and thus zero thermal Hall effect. The authors need to explain why the unconventional thermal Hall effect could happen with the SU(2) effect. (one may think about the anomalous Hall effect in ferromagnetic and antiferromagnetic systems.)
3. The referee is aware of one previous work on ferromagnetic skyrmion GaV₄Se₈, where, as the authors pointed out, inconsistency among theories exists. The authors should make it clear how this paper's approach can address this issue to avoid ambiguous results from simulations.

Reviewer #2 (Remarks to the Author):

In the manuscript, the author studied the anomalous transport caused by magnon excitation in insulating ferro- and antiferromagnets. This is like the anomalous Hall effect of conducting electrons. The thermal Hall coefficient takes a substantial value when the material enters a three-sublattice antiferromagnetic skyrmion phase, which is confirmed by the large-scale spin wave theory. The study is comprehensive, and the discussion is reasonable. But I have some concerns about the measurement and the data showing. Also, the language needs to be improved. Before that, I cannot recommend for the publication of this paper in Nature Communications.

I will list my concerns below:

1. In Figure 1(a), there are two planes. One is in grey and the other in blue. The caption says the blue one is [111]. What about the grey one?
2. In the supplementary material, the author shows the setup for thermal conductivity measurement and the size of the real crystal. How does the thermal pulse look like, or in other word how is the profile of temperature difference between the hot and cold sides? I'm curious about this because for a 1mm crystal, the thermal insulating between the hot and cold sides is important to get a good temperature difference for thermal conductivity measurement.
3. In Figure 1(d), the field dependence of the thermal conductivity is about 0.1 (10%) around peak and dip. What is the error bar for thermal conductivity measurement? The error of thermal conductivity measurement sometime could be comparable to that order.
4. In Fig 2, the main text states that the a-d should be magnon bands. But the caption says 'energy bands'. Could the author keep them uniform? Or are they different?
5. In Fig 3, there are error bars for the κ_{xy} but no error bar for the κ_{xx} . As I said in Q3, the error bar is important especially for the study of the field dependence as the error itself may be in the order of the field dependence.
6. Some typos like in page 2 '...forming a triangular lattice and stacks...' should be 'forming... and stacking...'.

Reviewer #3 (Remarks to the Author):

The manuscript reports measurements of the low-temperature thermal transport in magnetic field up to 14T on single crystals of spinel antiferromagnet MnSc_2S_4 . The observed magnetic field dependence of thermal transport and thermal Hall effect suggest dominant contribution of magnetic excitations, the spin waves, to the heat transport in the ordered state below $T_N = 2.3\text{K}$. The experimental results are adequately explained by classical spin-wave theory calculations for a large magnetic unit cell accounting for the non-collinear three-sublattice antiferromagnetic skyrmion-lattice magnetic structure of this material and by the evolution of this magnetic structure in magnetic field. From the analysis of magnetic field, temperature, and sample dependences of the measured heat transport, authors also infer the low-temperature B-T phase diagram. I think that the experimental data reported in the manuscript are measured reliably and that the spin-wave theory explanations are plausible and likely correct. However, the fact that the Hall thermal conductivity measured on two different samples differs by an order of magnitude (and diagonal thermal conductivity is vastly, ~ 3 times different, too) leaves me with a serious suspicion that this could either be an extrinsic property of the material, or perhaps a measurement artifact. Authors need to present some convincing argument that would rule this out and explain such vast difference between the two samples. But even leaving this suspicion aside, I do not find the results reported in the manuscript to be of sufficient importance and broad interest for researchers to recommend publication in Nature Communications. I rather think that these results are suitable for publication in a focused specialist journal such as Physical Review B. Having said the above, I would also like to add that the title claim of the manuscript ("Emergent $\text{SU}(3)$ magnons"), as well as a number of conclusions which authors report as central findings are misleading, and, to a large extent misrepresent the results. The manuscript also abounds with terminology that is used as hollow, meaningless buzzwords, such as "classical spiral spin liquid", "spinons" which have no relation to the reported data and in fact are misleading and distracting for the reader.

Specifically, the transformation to a rotating frame is a standard, textbook approach of spin-wave theory for helimagnets and other non-collinear structures. That it can be recast in the gauge field language is also well-known. It is important to recognize that using this gauge field language is not mandatory for calculating spin waves and their field-dependent contribution to heat transport – this can be straightforwardly done using spin-wave theory calculation and without using gauge field language. It can probably also be done in a number of other ways, too. That for a non-Bravais multi-sublattice spin system, such as the material authors study, different rotations are needed for different sublattices in order to align spins with the quantization axes prior to recasting Hamiltonian in boson language via Holstein-Primakoff transformation is also well-known. The diagonalization of the ensuing matrix for obtaining the spin-wave spectrum is a common textbook procedure. Unsurprisingly, when this procedure for a 3-sublattice system is recast into gauge field language, the resulting gauge field is $\text{SU}(3)$. Authors report this result as "emergent $\text{SU}(3)$ magnons", which I find a misnomer. Firstly, this $\text{SU}(3)$ gauge field is not a property of the physical system under study, but a property of a particular description of that system. A number of other descriptions are possible, including traditional spin-wave theory, which can reproduce the observed results without using the gauge field language. Of course, authors are free to choose any language they like – gauge field, slave fermions or bosons, or whatever. But finding that in a specific (gauge field) language spin waves in the system can be described as " $\text{SU}(3)$ magnons" does not amount to their "emergence" outside of this specific description. And their existence in this specific description does not (in my view) amount to a significant discovery. Secondly, this appears to be a misuse of the "emergent" terminology: these magnons are not emergent as per conventional usage [P. W. Anderson, Science 177, 393 (1972)].

Below, I list a number of other comments, which in my opinion could help authors in revising their manuscript for submission to a specialized journal.

Introduction

A number of passages in the abstract, introduction, and throughout the manuscript, read such as if authors render physical reality to theoretical constructs used for describing physical phenomena in a material systems. While in an exact solution of Kitaev model spin-1/2 excitations are indeed described in terms of Majorana fermions and gauge fluxes, this is a property of specific theoretical description, not the underlying physics. Other theoretical descriptions are also possible - whether or not they would lead to exact results, but that might be just a technical problem. When a spin-1/2 is represented using slave bosons, or slave fermions, or a fermion with a gauge field as in Jordan-Wigner-type transformation, the resulting objects are property of the corresponding theoretical description, not of the material system to which that description applies. Same material system can be described via different theories using bosons, fermions, anyons, with gauge fields or without, etc. The appearance of these useful auxiliary entities in the corresponding description does not imply that they actually exist outside of that description, in the real material. In order to not mislead the reader this point should be made abundantly clear throughout the manuscript.

Page 1, first paragraph. "... Hall conductivity reported in α -RuCl₃ is argued as being carried by these fractionalized Majorana fermions"

Can be theoretically described as being carried by these (auxiliary) Majorana fermions

Page 1, second paragraph. ... is explained by the gauge invariant referred to as a Chern number" It would be better to say "... is described by the gauge invariant referred to as a Chern number", which makes it explicitly to be a property of the theoretical description.

Page 1, third paragraph. "Magnons in insulating magnets are simple bosonic excitations but can also carry a U(1) gauge; ..."

Magnons in insulating magnets not "are" simple bosonic excitations - they can be described as such. They can also be described as interacting fermions with gauge fields attached, using slave fermions or slave bosons, and probably in many other ways. In some cases, like the one authors consider, it is useful to attach a gauge field to bosons (or fermions) on which spin operators are mapped. It needs to be clearly communicated that this gauge field is a property of specific theoretical description and not of the system itself. In general, unless discussion makes clear that "explained" refers to theoretical model, "described" should be used instead.

Similarly, "report the topological transport of magnons due to U(1) gauge field" should be rephrased as "report the topological transport of magnons which can be described by U(1) gauge field"; "thermal Hall effect is explained by the U(1) gauge fields" -> "thermal Hall effect is described by the U(1) gauge fields".

Page 1, 4th paragraph. "a more abundant gauge structure may appear successively in the transport phenomena." -> "a more abundant gauge structure may appear useful in descriptions of the transport phenomena."

"the U(1) gauge picture is not applicable" -> the U(1) gauge picture is not useful

"using the SU(2) gauge field to have the anomalous thermal Hall effect" -> "using the SU(2) gauge field to describe the anomalous thermal Hall effect"

Page 1, 5th paragraph. "By performing a large-scale period spin-wave theory show that the carriers are the magnons in a complex SU(3) gauge field originating from the significant three-sublattice structure"

The "large-scale period" terminology is unclear and, frankly, awkward. Does it mean "by performing large unit cell spin-wave theory calculations"?

I would suggest rephrasing the passage to something like "Our large unit cell spin wave theory calculations show that the heat carriers can be described as the magnons in a complex SU(3) gauge field originating from the three sublattice structure" to make clear that the U3 gauge field is the property of the specific theoretical description that authors use (and also abandon "significant", which does not make much sense in the context)

Page 4, Eq.(2) . The lattice to which BZ where integration is carried over corresponds needs to be specified

Page 4. "... χ_{xy} show peaks at 5 and 7 T ..." -> ... calculated χ_{xy} has peaks at 5 and 7 T ...

Page 5, top paragraph, and elsewhere. "we conclude that a magnetic excitation possibly related to the spiral spin liquid phase"

There is a repeated attribution of observations that do not agree with authors' spin wave theory analysis to the putative "spiral spin liquid phase". Firstly, this is not substantiated by the presented data. Secondly, the terminology of "spiral spin liquid" to describe a correlated frustrated or quasi-low-dimensional paramagnet where at finite temperature there is a manifold of thermally excited states with specific diffuse signature in Q-space, which is something well-known, is quite misleading in itself.

Page 5, discussion, first paragraph. "To understand these features, we calculate how many magnon bands from the bottom would give major contributions to χ_{xy} ..."

This choice is arbitrary and unphysical. The Brillouin zone size is changing with change of superlattice unit cell and the corresponding change in the number of magnon bands. What authors should investigate instead, is not dependence on the number of magnon bands but on the energy window cutoff (which also truncates the number of bands, but is a physical quantity which is invariant with respect to change of the size of the Brillouin zone).

Supplementary

>"in frustrated insulating quantum magnets, such strong spin correlation, even though they may not form a long-range order, is known to sometimes host a spinons excitation that carries heat current" Speculations on spinons are esoteric and read out of the context here. Importantly, they also are not supported by the data. In a correlated paramagnet such as the present system where TN is much less than Curie-Weiss temperature, paramagnons are present above TN (and below the curie-Weiss temperature) and, even though short-lived, can still contribute to thermal transport. Furthermore, I do not see how "classical spiral spin terminology" is useful for describing the correlated paramagnet - on the contrary, it looks to be harmful as it lead present authors to unsubstantiated speculations about some hypothetical "spinons" in this system, which is unphysical, unuseful, and unsubstantiated by any of the data presented in the manuscript. Overall, this does look like a hollow buzzword usage, which should be avoided.

>"Since magnetic skyrmions are a metastable magnetic order"

Is it indeed metastable? Or, is this only the case in the coexistence region of the first order phase transition?

>"Although the amplitude of χ_{xy}/T of sample #2 is about 1/10 of that of sample #1, "

If measurements of the same quantity on two different samples of the same substance yield results that differ by an order of magnitude, there is a serious suspicion that the quantity is extrinsic. It is further concerning that different scaling needs to be used for k_{xx} (x3.2) and k_{xy} (x10). Authors need to present some convincing argument that would rule this out.

>"The peak at around 4 T and the dips at 7 T are the anomalies that take place at the magnetic phase transition points reported previously."

What is the rationale for associating "peak" or "dip" with a phase transition? Conventionally, phase transition of n-th order is manifested by a discontinuity of an (n-1)-th derivative of a thermodynamic potential. For the first order phase transition, thermodynamic potential itself is discontinuous, for the second order it has a cusp, such as usually observed in magnetic susceptibility (adding $\frac{1}{2}\chi H^2$ to free energy) at a second order phase transition. What is the scenario (what type of singularity and in what quantity) authors envision for their "peaks" and "dips" and what type of magnetic phase transition (if any) does it correspond to?

>the field angle may influence the structure of the domain and accordingly, the magnitude of orderings.

What "magnitude of orderings" do authors mean here? I think they argued that this is a classical-spin system and local moments are more or less in a fully saturated state at low T?

>the emergent AFM-SkL phase reported previously

In what sense is this phase "emergent"? It looks like a simple classical ground state for this lattice spin Hamiltonian?

>Eq S3 and elsewhere

Indexing needs to be explained (ie, in what lattice $(1, 1, 1)$, $(1, 1, 0)$, etc, are indexed?).

> $q = 3\pi/2 (1, 1, 0)$ reciprocal lattice vectors are $b_1 = \pi/2 (1, 1, 0)$, $b_2 = 2\pi(1, -1, 0)$, $b_3 = 2\pi(0, 0, 1)$.

Relation between propagation vector of the magnetic structure and need to be shown here and elsewhere. In the above, it appears that $q=3b_1$ - why factor 3 is needed here? The wave vector equal to integer number of reciprocal lattice units is same as $q=0$? Perhaps, authors have something in mind here, but it is difficult to understand what that is.

> $q_1 = 3\pi/2 (1, -1, 0)$, $q_2 = 3\pi/2 (1, 0, -1)$, $q_3 = 3\pi/2 (0, 1, -1)$

See comment above.

>Eq. S12

All symbols in the equations need to be defined.

>artificially setting $J_{\parallel} = 0$, which gives the coplanar magnetic structure instead of the helical one
Helical configuration can be coplanar. It is unclear what is meant here. Conical? Simply non-coplanar?

>This indicates that the anisotropic exchange interactions

This is somewhat misleading, as anisotropy derives from spin-orbit interactions. Perhaps, just saying "anisotropic interactions" might be better.

>Eq. S13

Here, the indexing of the wave vector is in the units of small reciprocal lattice corresponding to large real-space unit cell. On the other hand, indexing in the paramagnetic crystal lattice is used in Eqs S4 and S5. This needs to be properly explained and the relation between these indexing established. Also, the relation of the two expressions shown needs to be described (the second line looks like a second-order perturbation theory?)

>Eq. S14

All quantities (symbols) need to be described

>Eqs. S15 - S19

Indexing needs to be specified (in which lattice r and q_m , Q_m , are indexed? Do they refer to continuous space, Angstroms and Angstroms inverse? From Eqs. S16, S17, it appears they refer to a specific reciprocal lattice - explain which one)

>The $U(1)$ gauge field is constructed using slowly varying sets of $m_l(r)$.

From the rotation matrix determining the direction of local quantization axes?

Response to the remarks by Reviewer #1

We thank Reviewer #1 for the nice summary of our work and for constructive comments. For all the suggestions 1-3 given, we made revisions together with the point-by-point reply in the following.

Reviewer #1: *1. The authors point out the importance of symmetry, U(1), SU(2), and SU(3), in describing the excitation of solid-state systems. However, there is no schematic illustration for these symmetries and how they are linked to the magnon in this work. The referee suggests that the authors need to draw illustrations to highlight this link.*

Reply: Thank you for the important suggestion. We add new Fig. 4(b)-(d) to give the schematic illustration of gauge fields/fluxes and how they are linked to magnons. We agree that there had been a gap in the explanation between Fig.4 and the gauge-field picture realized also in other systems.

To be brief, the U(1) gauge field is known NOT to afford the thermal Hall effect in the bipartite lattices (e.g. square lattice antiferromagnets) because of the no-go theorem. However, when the higher-rank SU(2) or SU(3) gauge fields appear, its non-commutative nature generates a fictitious higher-rank flux and breaks the time reversal symmetry. This gives room for the bipartite antiferromagnets to form a thermal Hall effect, which is observed in the present paper experimentally for the first time.

Please notice that the SU(2) or SU(3) gauge field does not always/necessarily appear for two-sublattice or three-sublattice magnetically ordered states. It depends on the details of the Hamiltonian and the magnetic structure of the state.

Below, let us expand it in more detail.

(i) The excitation of ferromagnets is described by a single species of magnon, and due to DM interactions or non-coplaner magnetic orderings (including ferro skyrmions), the hopping term of magnon has the coefficient $U_{ij} = e^{i\theta_{ij}}$ (Peierls phase) taking the form of U(1) vector potential, which is the source of the Berry curvature. The gauge-invariant quantity is the flux ϕ defined on the closed loop(plaquette) as, $U_{12}U_{23}U_{34}U_{41} = e^{i\phi}$. However, for the edge shared square lattice and triangular lattice, the contribution from fluxes ϕ and $-\phi$ that appears on the neighboring plaquettes (see new Fig.4(b)) are cancelled out due to the lattice symmetry combined with time reversal symmetry and gives zero thermal Hall effect. In Ref.[19] it is explained as the no-go theorem.

(ii) The excitations of two-sublattice antiferromagnets are described by two species of

magnons a_i, b_j defined as bosons on two sublattices A and B. In the similar situation as (i), two U(1) gauge fields can be independently generated for a_i, b_j . When they are decoupled, the two U(1) gauge fields suffer no-go rule of cancellation.

If the two U(1) gauge fields are coupled, they form a non-commutative SU(2) gauge field rerepresented by $T_{ij} = e^{i\theta_{ij}}$ where θ_{ij} is the 2x2 unitary matrix.

Due to the non-commutativity we find $T_{12}T_{23}T_{34}T_{41} = e^{i\Phi\cdot\sigma}$ and $(T_{12}T_{23}T_{3'4'}T_{4'1})^+ = e^{i\Phi\cdot\sigma}$ for the adjacent plaquettes (which has nearly the same sign), namely the system has a uniform flux configuration breaking the time-reversal symmetry, and the cancellation no longer occurs. Therefore, finding an interaction or magnetic profile that couples the two U(1) gauge fields equals finding a system with thermal Hall conductivity.

(iii) In the present three-sublattice skyrmion state, the natural description of its magnetic excitations is given by using the three species of magnons, which naturally couple and form a non-commutative SU(3) gauge field.

The simplest explanation (formulation) on the way how the SU(3) gauge field appears is written in the Discussion. The emergent SU(3) gauge field is the first and the only natural explanation on the numerical(analytical)result: i.e. the precise spin-wave calculation in the present case, that allows the thermal Hall effect in the edge-shared lattice which were forbidden by the no-go picture.

Reviewer #1: *2. The authors point out that the antiferromagnetic skyrmion phase supports SU(3) symmetry, which is in contrast to U(1) symmetry in the ferromagnetic skyrmion phase. In general, antiferromagnetic skyrmion should have degenerate magnon bands and thus zero thermal Hall effect. The authors need to explain why the unconventional thermal Hall effect could happen with the SU(2) effect. (one may think about the anomalous Hall effect in ferromagnetic and antiferromagnetic systems.)*

Reply: Firstly, the comment “*In general, antiferromagnetic skyrmion should have degenerate magnon bands*”, is not true. To our knowledge, the present skyrmion is the only antiferromagnetic skyrmion experimentally found so far, and there are no degenerate magnon bands within the spin-wave theory. We clarify it in the text.

We suppose Reviewer #2 has in mind a typical linear and gapless degenerate magnon bands of the two-sublattice antiferromagnet, which indeed show zero thermal Hall effect.

The bipartite antiferromagnets that show finite thermal Hall effect (see Ref.[27] for theory) no longer shows magnon-band degeneracies due to the magnetic anisotropies or DM

interactions. These effects are essential to generate SU(2) gauge field from two U(1) gauge field.

We hope the revisions of manuscript related to comment-1 and -2 will suffice the inquiry about the clarification of the origins of thermal Hall effect.

Reviewer #1: 3. *The referee is aware of one previous work on ferromagnetic skyrmion GaV₄Se₈, where, as the authors pointed out, inconsistency among theories exists. The authors should make it clear how this paper's approach can address this issue to avoid ambiguous results from simulations.*

Reply: Thank you for your interest on the ferromagnetic skyrmion GaV₄Se₈, which part of the authors are involved in.

The inconsistency comes from the oversimplification of the U(1) gauge theory adopting the simple hopping Hamiltonian of bosons with the U(1) Peierls phase. It is derived from the Heisenberg Hamiltonian by dropping off the terms of higher order variation of magnetic moments in space; the dropped terms include particle-non-conserving terms such as $b_i b_j$ or $b_i^+ b_j^+$ which should have essentially influenced the Berry curvature of the low-energy magnon bands if present. The spin wave theory includes the effect of all magnetic terms in the original spin model, and are more quantitatively reliable and can compare with experiments. However, we are not intended to deny the benefit of field theories because they give rough physical picture of what kind of effects or spin configurations are important to have a thermal Hall effect.

We revised the 2nd paragraph of the Discussion.

Response to the remarks by Reviewer #2

We thank Reviewer #2 for carefully reading the manuscript and for the positive assessments noting that *"the study is comprehensive, and the discussion is reasonable"*.

Regarding the comment on "some concerns about the measurement and the data showing", we have responded to all the issues addressed, corrected the way of data showing, and improved the language to our best.

We make point-by-point replies to the comments in the following.

Reviewer #2: 1. *In Figure 1(a), there are two planes. One is in grey and the other in blue.*

The caption says the blue one is [111]. What about the grey one?

Reply: We apologize for the confusion. In Fig.1(a) we wrote the two planes both in blue (but are written transparently). These planes are parallel with each other and are perpendicular to the [111] direction. The difference between the two are that they enclose different sublattice sites of the diamond lattice formed by Mn^{2+} ions. Namely, we need these two planes to define the unit cell. We revised the caption in our revised manuscript for clarification.

Reviewer #2: *2. In the supplementary material, the author shows the setup for thermal conductivity measurement and the size of the real crystal. How does the thermal pulse look like, or in other word how is the profile of temperature difference between the hot and cold sides? I'm curious about this because for a 1mm crystal, the thermal insulating between the hot and cold sides is important to get a good temperature difference for thermal conductivity measurement.*

Reply: Thank you for the question. Indeed, we need to clarify that the temperature slope for our crystal size is stable to guarantee the quality of our measurement.

We added a new figure in our Supplementary materials (Fig. S2, shown below). A stable temperature gradient (ΔT_x was typically set to 3–5 % of the sample temperature) was well established during the measurement despite the small size of the sample. After waiting for the stabilization of the sample temperature, we averaged the data for 240–300 secs for each measurement. The stability of the sample temperature during the period is typically 0.002% of the sample temperature, which is small enough to resolve ΔT_y^{asym} of our measurements.

A representative time profile of the sample temperature ($\frac{T_{\text{High}} + T_{\text{L1}}}{2}$, left axis), $\Delta T_x \equiv T_{\text{High}} - T_{\text{L1}}$, and $\Delta T_y \equiv T_{\text{L1}} - T_{\text{L2}}$ (right axis) at 0.5 K and 6.5 T. A heat current $Q = 35$ pW was turned on at the time shown by the dashed line.

The heater and the thermometers used in our thermal-transport measurements were thermally well isolated from the LiF heat bath by placing them on thin Kapton tubes and using resistive manganin wires for the electric connections. The thermal resistance between the thermometers and the heat bath is at least two orders magnitude larger than that of the sample in the whole temperature range of our study.

We added the above figure and descriptions in SM.

Reviewer #2: 3. In Figure 1(d), the field dependence of the thermal conductivity is about 0.1 (10%) around peak and dip. What is the error bar for thermal conductivity measurement? The error of thermal conductivity measurement sometime could be comparable to that order.

Reply: The error bars of each κ_{xx} and κ_{xy} measurements can be estimated from the error in determining the temperature differences, ΔT_x and ΔT_y . As shown in the figure above, the error in estimating the temperature difference of about 0.03% results in the error in κ_{xx} less than 0.1%, which is smaller than the symbol size. There is also a systematic error of up to about 1% between measurements done under almost the same conduction (field-up and field-down measurements at the same temperature) for the lowest temperature data, which is much smaller than the measured magneto thermal conductivity as shown below.

We added in Fig. 1d these missing error bars in our revised manuscript.

Revised Fig. 1d with error bars estimated from the difference in the field-up and field-down measurements at the same temperature. Note that the error bars are larger than the symbol size only for the low temperature data.

The reviewer might concern the uncertainty in determining the absolute value of κ_{xx} for the large error in thermal conductivity measurements. As we have described in our SM, there is a large ambiguity in determining the geometrical factors (width, length, and the thickness of the sample) because of the irregular shape of the sample and the finite size of the thermal contacts (see Fig. S1(b)). This ambiguity results in the uncertainty in estimating the absolute value of the magnitude of κ_{xx} and that of κ_{xy} by a factor of 2–4. We note that this ambiguity only brings a factor change in the measured quantity, thus canceled out in estimating the magnetothermal conductivity ($\Delta\kappa_{xx}(B)/\kappa_{xx}(0)$).

Reviewer #2: 4. *In Fig 2, the main text states that the a-d should be magnon bands. But the caption says 'energy bands'. Could the author keep them uniform? Or are they different?*

Reply: Thank you for pointing out. We revised all the “energy bands” to “magnon bands” which is more precise.

Reviewer #2: 5. *In Fig 3, there are error bars for the kappa_xy but no error bar for the kappa_xx. As I said in Q3, the error bar is important especially for the study of the field dependence as the error itself may be in the order of the field dependence.*

Reply: Thank you for the suggestion. Without intention we missed putting error bars in Figs.1d and S4. In the updated figure we add them in the same way as we have evaluated the errorbars of κ_{xy} . (See also the reply to comment-3 given above).

Reviewer #2: *comment-6. Some typos like in page 2 '...forming a triangular lattice and stacks...' should be 'forming... and stacking...'.*

Reply: Thank you for pointing out. We corrected the typos including the ones addressed to our best in our revised manuscript by carefully rereading the manuscript.

Response to the remarks by Reviewer #3

We thank Reviewer #3 for giving proper summary of our work and for the comment, "*I think that the experimental data reported in the manuscript are measured reliably and that the spin-wave theory explanations are plausible and likely correct.*", supporting the reliability and consistency of our results between experiment and theory.

In the other part of the comments given, we find some difference or inconsistencies with Reviewer #3 in the idea of how the physical pictures in condensed matter physics/theories shall serve. We suspect that it comes from the misunderstandings on the role of the SU(3) gauge field and the existence of related gauge invariants. Since our previous manuscript lacked explanations on these points, we revised the Discussion section extensively.

We thank the Reviewer#3 for his/her effort of reading the manuscript including the supplementary in real detail, which was helpful to clarify the details and to improve the presentation of our manuscript.

[EXPERIMENTAL PART]

Reviewer #3: *However, the fact that the Hall thermal conductivity measured on two different samples differs by an order of magnitude (and diagonal thermal conductivity is vastly, ~ 3 times different, too) leaves me with a serious suspicion that this could either be an extrinsic property of the material, or perhaps a measurement artifact. Authors need to present some convincing argument that would rule this out and explain such vast difference between the two samples. ...*

"Although the amplitude of κ_{xy}/T of sample #2 is about 1/10 of that of sample #1, "If measurements of the same quantity on two different samples of the same substance yield results that differ by an order of magnitude, there is a serious suspicion that the quantity is extrinsic. It is further concerning that different scaling needs to be used for κ_{xx} (x3.2) and κ_{xy} (x10). Authors need to present some convincing argument that would rule this out.

Reply: Thank you for the comment. Despite the concerns raised, we first confirm that our main experimental result, the field-dependence in the upturn of thermal Hall conductivity, $\kappa_{xy}(B)$, and in the magnetothermal conductivity, $(\kappa_{xx}(B) - \kappa_{xx}(0))/\kappa_{xx}(0)$, presented in Fig.1 can scarcely occur as an extrinsic effect.

The main reason for Reviewer's concern is the absolute value of thermal transport coefficients κ_{xx} and κ_{xy} being different between the two samples. However, it is subtle (except for specific case like quantization) to give any physical meaning on the absolute value

of thermal transport coefficients. This is because thermal transport coefficients measured under non-equilibrium condition are always subject to scattering effects specific to each sample quality, and accordingly a large sample dependence of its magnitude is naturally observed and are often inevitable. This point is different from heat capacity and magnetic susceptibility measured in thermal equilibrium.

Unlike electronic systems, κ_{xx} consists of contributions from phonons and magnons, and the latter is responsible for a major part of the field (B) dependence. Although the two contributions at $B = 0$ is not clearly separable, the large difference in the magnitude of κ_{xx} of the two crystals (Fig.S6(a)) indicate a large difference in the contribution from phonons, which is largely influenced by the **phonon mean-free path (see Fig.S3) due to sample quality**. The κ_{xx} of sample #2 is about 1/3 of that of sample #1 (Fig. S6(a)), which indicates a shorter phonon mean-free path showing that phonons in sample #2 suffer stronger scattering effect.

The field dependent part $((\kappa_{xx}(B) - \kappa_{xx}(0))/\kappa_{xx}(0))$ in Fig. S6(b) of sample #2 scales with sample #1 by a factor of 3.2, indicating that the fraction of the magnetic component in sample #2 is about 1/3 of that of sample #1.

Combining these two data, we find that

- the magnetic component of κ_{xx} of sample #2 is about 1/9 of that of sample #1. This has almost the same fraction of κ_{xy} of sample #2 compared to that of sample #1 shown in Fig. S6(c). Therefore, the scaling factors consistently explain that the magnetic components in both samples are intrinsic ones. (This discussion was previously missing).

We further find that

- Fig. S6 (b,c) the field-dependence of the data, $(\kappa_{xx}(B) - \kappa_{xx}(0))/\kappa_{xx}(0)$ and $\kappa_{xy}(B)/T$ scale almost perfectly by multiplying aforementioned constant scaling factor. This cannot be any artifact or coincidence.

In various magnetic insulators the sample with smaller amplitude of κ_{xx} is found to have smaller fraction of the magnetic components (e.g. kagomé AFM volborthite, see Fig. 3 in M. Yamashita et al., J. Phys.: Condens. Matter 32 074001 (2020). and M. Akazawa et al., Phys. Rev. X 10, 041059 (2020).). Therefore it is natural to expect that the magnetic thermal conduction is strongly suppressed when the phonon thermal conduction is suppressed.

They all consistently point to the conclusion that **the field dependence is the intrinsic effect of magnons**, persisting in the different scattering environments, never be a measurement artifact. We do not think there is a more natural and logical interpretations on the data as “the artifacts”.

Since the above-mentioned discussions on the scaling factor is not fully presented in the main text nor in the SM, we added them. Thank you for pointing out.

Notes: Reviewer #3 may have in mind the discussions of intrinsic and extrinsic mechanisms for anomalous Hall effect (AHE) in ferromagnetic metals (electrons are the carriers).

However, it is for electrons, not for magnons, since the scattering mechanisms differ.

In the AHEs in ferromagnetic metals, the Hall conductivity (σ_{xy}) have both intrinsic mechanism (coming from Berry phase effects) and extrinsic mechanism (by either skew or side-jump scatterings). They show different relationships with the longitudinal conductivity (σ_{xx}): The σ_{xy} by the intrinsic and the side-jump mechanisms do not depend on σ_{xx} , whereas σ_{xy} by the skew scattering is proportional to σ_{xx} . However, even the intrinsic σ_{xy} is not completely free from the scattering effect. The scattering broadens the conduction bands and suppresses the intrinsic σ_{xy} . [see Y. Shiomi, Y. Onose, and Y. Tokura, Phys. Rev. B 81, 054414 (2010)].

Reviewer #3 might concern in analogy with above mentioned electronic case that our magnon κ_{xy} mechanism should be free from any scattering effects on magnons. The magnon-phonon scattering is indeed subtle at low energies. However, similarly to the electrons, the broadening effect also suppresses the intrinsic κ_{xy} given by our SU(3) gauge-field mechanism. The smaller κ_{xy} of sample #2 can be explained by a combination of the broadening effect on the thermal Hall mechanism and the suppression of the magnon contribution in κ_{xx} .

[THEORETICAL PART]

Let us first summarize the two major comment by Reviewer#3:

(#3-1) As in the Kitaev model where the *spin-1/2 excitations are indeed described in terms of Majorana fermions and gauge fluxes*, the emergent SU(3) gauge *“is a property of specific theoretical description, not the underlying physics. Other theoretical descriptions are also possible.”*

(#3-2) *“Finding that in a specific (gauge field) language spin waves in the system can be described as “SU(3) magnons” does not amount to their “emergence” outside of this specific description. And their existence in this specific description does not (in my view) amount to a significant discovery”.*

We first clarify that we do not agree with both of these comments. We first summarize the main point our reply here.

Regarding (#3-1), theorists can indeed choose whatever available descriptions or formulations they want in dealing with the phenomena. However, whether the phenomena is clearly understood to wide scientific communities or not depends crucially on the choice of

description. The Majorana fermions and gauge fluxes are the best successful description for Kitaev spin liquid state, as one can see that it is the most widely accepted. We agree that we find no reason to exclude the many body descriptions on the same topic, since one author indeed done the many-body calculation on the Kitaev model and reproduced the data reported from the Majorana-based calculation, see arXiv:2308.02015. However this does not mean that the Majorana+gauge picture is unimportant.

The natural choice of description has provided major achievements in physics, including Fermi liquid theory and magnon excitations. We do not agree with (#3-1) and argue that it has nothing to do with a neutral judgement on the paper.

Regarding (#3-2), we think that it is based on the misunderstanding of how we derived the SU(3) gauge picture. The comment that the SU(3) gauge field is found in “*a specific (gauge field) language spin waves*” is not true. Our main theoretical result in Fig.2 showing nonzero thermal Hall effect based on the spin-wave theory has nothing to do with the choice of the SU(3) gauge-field description. In any case, we did not use any gauge language here, but the same order of approximation using different basis gives the identical/qualitatively the same result.

The SU(3) gauge field is adopted based on the field theory as simplified interpretation to the rigid result in Fig.2. It seems that Reviewer #3 suspects that the redundancy of gauge field (the remaining freedom of gauge transformation) makes our picture fragile. However, the thermal Hall effect is NOT directly relevant to the SU(3) gauge field itself but to the flux coming from the SU(3) gauge field which is a gauge-invariant quantity. It is well known that gauge invariants are physically meaningful and deserves the direct connections to underlying physics.

We also add the remark that the present thermal Hall effect is the first to be observed experimentally in insulating antiferromagnets with more than two sublattices, and using the natural choice of description its origin is clarified using the SU(3) flux. In our view: this fact is enough significant.

In the following we make point-by-point detailed replies to the comments.

Reviewer #3: It is important to recognize that using this gauge field language is not mandatory for calculating spin waves and their field-dependent contribution to heat transport—this can be straightforwardly done using spin-wave theory calculation and without using gauge field language.” Specifically, the transformation to a rotating frame is a standard, textbook approach of spin-wave theory for helimagnets and other non-collinear structures. That it can be recast in the gauge field language is also well-known.

Reply: Indeed, the gauge field language is not mandatory for the spin-wave calculation and field-dependent transport. We understand this point very well.

We do not understand this comment because we did not use the gauge field language in calculating the magnon bands and field-dependence of the thermal Hall conductivity; the non-Bravais multi-sublattice system with the rotation of the local spin axis we applied to perform the linear spin-wave theory calculations is the common textbook procedure.

It seems that using the rotating frame has made the wrong impression that the spin-wave results depend on the choice of basis or procedure. **However, it is the well-known fact that physical quantities/observables like magnon bands and κ_{xy} are always invariant about the choice of the basis.** We remark that the gauge field description is given only in the Discussion part for the interpretation of Fig.2.

Reviewer #3: It can probably also be done in a number of other ways, too. That for a non-Bravais multi-sublattice spin system, such as the material authors study, different rotations are needed for different sublattices in order to align spins with the quantization axes prior to recasting Hamiltonian in boson language via Holstein-Primakoff transformation is also well-known. The diagonalization of the ensuing matrix for obtaining the spin-wave spectrum is a common textbook procedure.

Reply: We understand that other choices of descriptions/formulations are also possible. However, for example, if one uses the slave fermions+gauge fields picture for the low energy excitation of magnetically ordered state, the description becomes the strongly interacting many fermions, and it is extremely hard or almost impossible to gain the intrinsic picture or the insight on the origin of the phenomena. Nearly free bosons are by no doubt clearest and the best description.

In this context, there is a **natural quasi-particle description, which cannot be freely chosen, but is embedded in the system.** We think this to be the “underlying physics”. Examples are

- quasi-particle picture of Fermi liquids
- Bogoliubov quasiparticles in superconductivity
- composite fermions in fractional quantum Hall liquids
- magnon description of magnetic excitations
- Majorana fermion description of Kitaev model ground state

There is a small possibility to be updated in the future but they are the best ones so far. We do not understand what the Reviewer #3 consider as the “underlying physics”, for which he/she seems to claim as the one that all the physicists could accept regardless of their

background, not dependent on the choice of description. This can only be the raw data, either numerical calculation, exact analytical solution, or the experimental measurement. According to his/her comment the Reviewer #3 seems to exclude all the above milestone findings from the underlying physics.

Reviewer #3: Unsurprisingly, when this procedure for a 3-sublattice system is recast into gauge field language, the resulting gauge field is SU(3). Authors report this result as “emergent SU(3) magnons”, which I find a misnomer. Firstly, this SU(3) gauge field is not a property of the physical system under study, but a property of a particular description of that system. A number of other descriptions are possible, including traditional spin-wave theory, which can reproduce the observed results without using the gauge field language. Of course, authors are free to choose any language they like—gauge field, slave fermions or bosons, or whatever. But finding that in a specific (gauge field) language spin waves in the system can be described as “SU(3) magnons” does not amount to their “emergence” outside of this specific description. And their existence in this specific description does not (in my view) amount to a significant discovery. Secondly, this appears to be a misuse of the “emergent” terminology: these magnons are not emergent as per conventional usage [P. W. Anderson, Science 177, 393 (1972)].

Reply: This comment is not true, but is based on the misunderstanding. We apologize that regarding the point on why we need the SU(3) gauge-field description, our previous manuscript lacked sufficient explanation, which may have caused the issue. We revised the manuscript according to the following explanation.

First of all, we need to clarify that the choice of 2-sublattice/3-sublattice on a square/triangular lattice antiferromagnet recast into the gauge field language does not necessarily result in the SU(2)/SU(3) gauge field physics.

For example, in the square lattice antiferromagnet, for simple Heisenberg model or by adding spin anisotropy, the thermal Hall effect and related SU(2) gauge field and the gauge invariant fluxes are absent. We showed in the theoretical paper Ref.[27] that

- a magnetic field perpendicular to the plane
- Dzyaloshinskii-Moriya interaction (antisymmetric magnetic exchange)

are required to explicitly have SU(2) gauge structure in the hopping constant of two species of magnons (regarded as pseudo up/down spin particles). In analogy to the Rashba electrons with spin-orbit coupling, these two effects yield the “pseudo-spin-orbit coupling” in the form of the SU(2) gauge field and give nonzero thermal Hall effect. (However, it is not experimentally observed yet.)

Therefore, whether the intrinsic SU(2) gauge field appears or not depends on the details of the Hamiltonian and not on the two-sublattice description.

In the same manner, in the triangular lattice, for the 120-degree uniform antiferromagnetic orderings with no spatial spin modulations, the thermal Hall effect is zero, and the SU(3) gauge-field description in our formulation is explicitly absent.

In this context, the SU(3) gauge flux is “emergent”.

We suspect that the discussion on the redundancy of gauge field which we presented in the introduction have misled the Reviewer #3; the true physical quantity like thermal Hall coefficients are not dependent on the choice of gauge description. **The gauge invariant quantities are meaningful**, but our SU(3) gauge field itself is not gauge invariant, and it can take different forms due to gauge redundancy (while they keep their form as the SU(3) gauge field). Here, the gauge-invariant quantity can be constructed by the Wilson loop; we add the example in Fig.2(b) the Wilson loop operator for the square lattice consisting of the product of gauge fields around the plaquette.

For the U(1) gauge field, the Wilson loop is $U_{12}U_{23}U_{34}U_{41} = e^{i\phi}$, yielding the flux ϕ as gauge invariant. This flux generates the Berry curvature and gives nonzero-thermal Hall effect. Unfortunately, on a square lattice, the flux $-\phi$ on the adjacent plaquette is related to the ϕ flux by the time reversal and translation operation, and due to the invariance under this operation, the contribution of the flux cancels out and give zero thermal Hall effect (see Katsura *et.al.* Ref.[19]) .

However, for the SU(2) gauge field, the Wilson loop as 2x2 matrix form $T_{12}T_{23}T_{34}T_{41} = e^{i\Phi\cdot\sigma}$ and the gauge invariant 2x2 fluxes, $+|\Phi|$ and $-|\Phi|$ appear in the adjacent plaquettes due to non-commutativity of T_{ij} (see the main text for detailed calculation), which do not cancel out anymore. This flux breaks the time reversal symmetry and give nonzero thermal Hall effect. As mentioned above, the SU(2)/SU(3) gauge flux do not necessarily appear for antiferromagnets. **The effect of gauge field we discuss here appears/disappears NOT because of the choice of representation, and it reflects the intrinsic property of the physical system/Hamiltonian under study.**

It might be natural to find that some audiences are unfamiliar with the fact that **the ordered antiferromagnets are widely believed NOT to exhibit a thermal Hall effect**. Part of the authors have proposed the thermal Hall effect of square lattice antiferromagnets due to the SU(2) gauge field, but it not experimentally observed yet so far. **The present work is the first report of not only AFM-SkL but the multi-subattice antiferromagnet exhibiting thermal Hall effect both in theory and experiment consistently.**

We made major revisions on the Discussion part to clarify these points.

Reviewer #3: *A number of passages in the abstract, introduction, and throughout the manuscript, read such as if authors render physical reality to theoretical constructs used for describing physical phenomena in a material systems. ...*

The manuscript also abounds with terminology that is used as hollow, meaningless buzzwords, such as "classical spiral spin liquid", "spinons" which have no relation to the reported data and in fact are misleading and distracting for the reader.

I would also like to add that the title claim of the manuscript ("Emergent SU(3) magnons"), as well as a number of conclusions which authors report as central findings are misleading, and, to a large extent misrepresent the results.

Reply: We believe that we have clarified enough our standpoint with the legitimate grounds that the SU(3) gauge-field description provides the most natural understanding of the thermal Hall effect observed in the edge-shared antiferromagnet observed experimentally for the first time.

Despite Reviewer #3's criticism on the title the "emergent SU(3) gauge", it is emergent because such gauge structure does not exist in the original Hamiltonian. It emerges because the AFM-SkL appears as thermodynamic phase by the symmetry breaking. However, we modify this part to "emergent SU(3) flux" since the gauge field itself is not invariant but the flux is, which might lessen the Reviewer #3 's "misleading" impressions.

We would like to notice that we have previously prepared the introduction very carefully by trying to avoid uncertain usages of words. For example, "classical spiral-spin liquid phase" is the terminology which Ref.[25] adopted and not ours, but the phase they reported is a "classical spin liquid" which is widely accepted as disordered but classical spin state (which is clarified as strongly correlated phase and is not a simple paramagnet). It is not in the same category as quantum spin liquid which Kitaev and pyrochlore U(1) phase represents. Although we confess that we do not necessarily favor them, we have no intention to strongly exclude them here, while we avoided them at most in the revised draft.

We adopted excitation of integrable spin chain(Ref.[1]), Kitaev anyons (Ref.[3-7]) and quantum spin ice/spin liquid (Ref.[8]) which are the well-known examples that the gauges or higher rank symmetries give a good descriptions. We see that the Reveiwer #3 disagrees with these descriptions, but the introductions are prepared for wide audiences with different backgrounds and these examples can naturally be accepted and connected to our work.

In explaining this part in the introduction, we mentioned that "*Indeed, there are several other cases that effective theories explaining the low-energy excitations of interesting quantum*

phases are not the simple bosonic or quasi-particle ones but are those subject to gauge fields.", which safely excludes the concerns of Reviewer#3 that these examples are dealt not as one of the interpretations or descriptions.

In any case, we agree with the Reviewer#3 that the terminologies need to be carefully presented, and have checked the introduction again, adopted many of the comments addressed by Reviewer #3 as in the following part of the reply. We also made extra small revisions that may have ruffled the understanding of Reviewer #3. Thank you for the many advices.

----- Replies to comments on the presentation -----

[Presentation of the main manuscript]

Reviewer #3: Page 1, first paragraph. "... Hall conductivity reported in α -RuCl₃ is argued as being carried by these fractionalized Majorana fermions"

Can be theoretically described as being carried by these (auxiliary) Majorana fermions

Reply: We did not fully adopt this revision because "can be theoretically described" is too strongly supporting the Majorana picture in the experiment in our view. We used "argued" because it is simply an argument by the authors in the literature cited, where we urge the attention of audiences that they follow the Kitaev picture mentioned in the sentence just before this.

Reviewer #3: Page 1, second paragraph. ... is explained by the gauge invariant referred to as a Chern number"

It would be better to say "... is described by the gauge invariant referred to as a Chern number", which makes it explicitly to be a property of the theoretical description.

Reply: We did not adopt this revision because the gauge invariant is the physically meaningful property, not dependent on the "description". We would like to clearly classify the difference between gauge invariant quantities and representation-dependent gauge fields.

Reviewer #3: Page 1, third paragraph. "Magnons in insulating magnets are simple bosonic excitations but can also carry a U(1) gauge; ..."

Magnons in insulating magnets not "are" simple bosonic excitations - they can be described as such. They can also be described as interacting fermions with gauge fields attached, using slave fermions or slave bosons, and probably in many other ways. In some cases, like the one authors consider, it is useful to attach a gauge field to bosons (or fermions) on which spin operators are mapped. It needs to be clearly communicated that this gauge field is a property of specific theoretical description and not of the system itself. In general, unless

discussion makes clear that "explained" refers to theoretical model, "described" should be used instead.

Reply: We disagree. This expression we made, *"Magnons in insulating magnets are simple bosonic excitations but can also carry a U(1) gauge; ..."* is the definition of magnons and has no ambiguity. We are not saying that excitation of insulating magnets are magnons, namely not excluding other possibilities. As we argued previously, the gauge field is already mentioned carefully in the 2nd paragraph of the introduction as not an invariant but a redundant, expression-dependent quantity. We ask Reviewer #3 to understand our careful context.

Reviewer #3: *Page 1, 4th paragraph. "a more abundant gauge structure may appear successively in the transport phenomena." -> "a more abundant gauge structure may appear useful in descriptions of the transport phenomena."*

"the U(1) gauge picture is not applicable" -> the U(1) gauge picture is not useful

"using the SU(2) gauge field to have the anomalous thermal Hall effect" -> "using the SU(2) gauge field to describe the anomalous thermal Hall effect"

Reply: We adopt part of these changes which we agree to be reasonable. However, *"the U(1) gauge picture is not applicable"*, this is a scientific fact which does not need change.

Reviewer #3: *Page 1, 5th paragraph. "By performing a large-scale period spin-wave theory show that the carriers are the magnons in a complex SU(3) gauge field originating from the significant three-sublattice structure"*

The "large-scale period" terminology is unclear and, frankly, awkward. Does it mean "by performing large unit cell spin-wave theory calculations"?

I would suggest rephrasing the passage to something like "Our large unit cell spin wave theory calculations show that the heat carriers can be described as the magnons in a complex SU(3) gauge field originating from the three sublattice structure" to make clear that the U3 gauge field is the property of the specific theoretical description that authors use (and also abandon "significant", which does not make much sense in the context)

Reply: We like these expressions, thank you for the proposal.

Reviewer #3: *Page 4, Eq.(2) . The lattice to which BZ where integration is carried over corresponds needs to be specified*

Page 4. "... kxy show peaks at 5 and 7 T ..." -> ... calculated kxy has peaks at 5 and 7 T ...

Reply: We clarified these points.

Reviewer #3: Page 5, top paragraph, and elsewhere. “we conclude that a magnetic excitation possibly related to the spiral spin liquid phase”

There is a repeated attribution of observations that do not agree with authors’ spin wave theory analysis to the putative “spiral spin liquid phase”. Firstly, this is not substantiated by the presented data. Secondly, the terminology of “spiral spin liquid” to describe a correlated frustrated or quasi-low-dimensional paramagnet where at finite temperature there is a manifold of thermally excited states with specific diffuse signature in Q-space, which is something well-known, is quite misleading in itself.

Reply: We see from this comment that we need to clarify that this consideration is only a speculation given from the experiments and not related to the theoretical calculation of the paper. At the same time, we agree that the terminology of “spiral spin liquid” is not necessary to express the results reported previously in Ref.[25]. However, it is beyond our control. We tried to avoid this expression at our most by modifying this part to:

“Therefore, we speculate the origin of the negative magneto thermal conductivity to be the magnetic excitation from a experimentally reported correlated paramagnet at finite temperature consisting of a manifold of states showing specific diffuse signature in momentum space, which is beyond the present theoretical treatment.”

Reviewer #3: Page 5, discussion, first paragraph. “To understand these features, we calculate how many magnon bands from the bottom would give major contributions to κ_{xy} ...” This choice is arbitrary and unphysical. The Brillouin zone size is changing with change of superlattice unit cell and the corresponding change in the number of magnon bands. What authors should investigate instead, is not dependence on the number of magnon bands but on the energy window cutoff (which also truncates the number of bands, but is a physical quantity which is invariant with respect to change of the size of the Brillouin zone).

Reply: This comment is totally based on misunderstanding.

Here, the energy window cutoff does not shrink the unit cell size. We are not checking the variation of the unit cell size (change of the ground state magnetic ordering) nor the Brillouin zone. This is meaningless for the present study because the spin configuration is fixed. We are not testing the variation of the ground state nor its stability which is already established in the previous works.

In Eq.(2) the κ_{xy} sums up the Berry curvature from $n = 1 \sim N_s$ magnon bands multiplied by the Bose distribution function. We simply demonstrated the weight of the lowest few bands, $N_s = 8,16,24$ to κ_{xy} in the evaluation of κ_{xy} (Fig.2(e)) and nothing more. The weight

changes with temperature based on the Bose distribution function and magnon-band density of states.

It is usually wrongly recognized that the majority of weight to κ_{xy} comes from the energy range of up to temperature. However, we showed that the weight of Bose distribution function particularly for the present dense band structure has significant contribution from energy much higher than the temperature. The data in Fig.S8 has no real physical meaning, it ensures which part of the magnon bands are important for κ_{xy} .

We changed the expression in the first paragraph of Discussion to avoid misunderstanding.

[Presentation of the Supplementary]

Reviewer #3: *"in frustrated insulating quantum magnets, such strong spin correlation, even though they may not form a long-range order, is known to sometimes host a spinons excitation that carries heat current"*

Speculations on spinons are esoteric and read out of the context here. Importantly, they also are not supported by the data. In a correlated paramagnet such as the present system where T_N is much less than Curie-Weiss temperature, paramagnons are present above T_N (and below the curie-Weiss temperature) and, even though short-lived, can still contribute to thermal transport. Furthermore, I do not see how "classical spiral spin terminology" is useful for describing the correlated paramagnet - on the contrary, it looks to be harmful as it lead present authors to unsubstantiated speculations about some hypothetical "spinons" in this system, which is unphysical, unuseful, and unsubstantiated bty any of the data presented in the manuscript. Overall, this does look like a hollow buzzword usage, which should be avoided.

Reply: We agree that discussing directly unrelated spinons might be harmful. We would like to simply point out the unbiased facts;

"In MnS_2S_4 , in a paramagnetic phase at low temperatures above T_N , the magnetic diffuse scattering experiments show the existence of some sort of magnetic excitations relevant to a manifold of wave numbers forming surfaces in the reciprocal space. This clear magnetic feature indicating a correlated paramagnet has a room to couple to magnetic fields and we speculate it to be the origin of the negative field dependence of κ_{xy}^{mag} . While this scenario is out of scope of the present theoretical treatment, ... "

Reviewer #3: *"Since magnetic skyrmions are a metastable magnetic order"*

Is it indeed metastable? Or, is this only the case in the coexistence region of the first order phase transition?

Reply: This expression was misleading. The skyrmions are only thermodynamically stable

namely, in a field and at finite temperature, meaning that it is not a ground state. We clarified this part. Thank you for pointing out.

Reviewer #3: >“The peak at around 4 T and the dips at 7 T are the anomalies that take place at the magnetic phase transition points reported previously.”

What is the rationale for associating “peak” or “dip” with a phase transition? Conventionally, phase transition of n-th order is manifested by a discontinuity of an (n-1)-th derivative of a thermodynamic potential. For the first order phase transition, thermodynamic potential itself is discontinuous, for the second order it has a cusp, such as usually observed in magnetic susceptibility (adding χH^2 to free energy) at a second order phase transition. What is the scenario (what type of singularity and in what quantity) authors envision for their “peaks” and “dips” and what type of magnetic phase transition (if any) does it correspond to?

Reply: In referring to the phase transition as cusps or discontinuities, Reviewer#3 is considering the thermodynamic quantities measured in thermal equilibrium. As we mentioned earlier, the thermal-transport measurements is given in a non-equilibrium and probe the entropy flow in the material. It is natural that the jumps or kink may not be expected experimentally in our data.

The rational reason for associating the peaks and dips is that they ALWAYS take place within a finite/limited resolution of the data at all phase transition points already reported. The previous report is based on different experimental probes and the phase boundaries are established by the comparison with theory. We are not trying to prove the existence of the transition. It is not realistic nor reasonable to distinguish the types of transition by our measurements.

Reviewer #3: >the field angle may influence the structure of the domain and accordingly, the magnitude of orderings.

What “magnitude of orderings” do authors mean here? I think they argued that this is a classical-spin system and local moments are more or less in a fully saturated state at low T?

Reply: Here, we refer to the “magnitude of orderings” as the average values of order parameters/magnetic structures in a field in the presence of multi-domains. According to Ref. [24] (S. Gao et al., Nature 586, 37–41 (2020)), the high-field fan phase in B // [111] has a multi-domain structure. Each domain has single-**Q** phases and there are six equivalent **Q**-vectors. If the magnetic field is tilted from [111] because six **Q**-vectors are no longer equivalent, and some of the domains are more preferred, i.e. the domain size distribution will change. At above 9T, the two samples #1 and #2 are expected to have different domain

distributions, namely the average magnetic structure differ. It is natural that they influence the amplitude of magnetothermal conductivity.

We add the explanation and modify the text as:

“In a high-field phase for $B \parallel [111]$ the multi-domain structure consisting each of single-Q, having six equivalent(different) Q-vectors is reported. If the magnetic field is tilted from [111], the six domains are no longer equivalent and their distribution change, namely the average magnetic structure will change. Under such effect, it is natural to expect different amplitudes of magnetothermal conductivity in the two samples.”

Reviewer #3: *>the emergent AFM-SkL phase reported previously*

In what sense is this phase “emergent”? It looks like a simple classical ground state for this lattice spin Hamiltonian?

Reply: Following the suggestion we deleted “emergent”. Well, we may say that it is a classical ground state for a toy model, but is not simple at all, nor trivial to be found in actual materials. How we see the phenomena depends on personal opinion.

Reviewer #3: *>Eq S3 and elsewhere Indexing needs to be explained (ie, in what lattice (1, 1, 1), (1, 1, 0), etc, are indexed?).*

Reply: These are components of vectors, not indices. We have already explicitly explained these details in the previous sentences, please take a look.

Reviewer #3:*> $q = 3\pi/2 (1, 1, 0)$ reciprocal lattice vectors are $b_1 = \pi/2 (1, 1, 0)$, $b_2 = 2\pi(1, 1, 0)$, $b_3 = 2\pi(0, 0, 1)$.*

Relation between propagation vector of the magnetic structure and need to be shown here and elsewhere. In the above, it appears that $q=3b_1$? why factor 3 is needed here? The wave vector equal to integer number of reciprocal lattice units is same as $q=0$? Perhaps, authors have something in mind here, but it is difficult to understand what that is.

> $q_1 = 3\pi/2 (1, 1, 0)$, $q_2 = 3\pi/2 (1, 0, 1)$, $q_3 = 3\pi/2 (0, 1, 1)$

See comment above.

Reply: We first clarify that the propagation vector \mathbf{q} is not necessarily equal to the ordering vector/ reciprocal lattice vector: \mathbf{q} can be multiples of the reciprocal lattice vectors. Also, the propagation vector $\mathbf{q} \neq 0$ is equal to $\mathbf{q} = 0$ in the folded Brillouin zone. While we believe that they are common knowledge in condensed matter physics, we explicitly notice it in the revised text.

Simplest example are:

* 1D system with a magnetic order characterized by $q = 3\pi/2$. Since the period of order is 4 site, the reciprocal lattice vector is $b = \pi/2$.

* In the square lattice antiferromagnet with $q = (\pi, \pi)$ is identical to (0,0) in the folded Brillouin zone.

Reviewer #3: *Eq. S12 All symbols in the equations need to be defined.*

Reply: Thank you for pointing out. We add the explanation of Xi and Delta.

Reviewer #3: *artificially setting $J_{\square} = 0$, which gives the coplanar magnetic structure instead of the helical one Helical configuration can be coplanar. It is unclear what is meant here. Conical? Simply non-coplanar?*

Reply: These are typo's. We apologize for the confusion. We correct them to helical.

Reviewer #3: *This indicates that the anisotropic exchange interactions This is somewhat misleading, as anisotropy derives from spin-orbit interactions. Perhaps, just saying "anisotropic interactions" might be better.*

Reply: "Anisotropic exchange interaction" is a common terminology of quantum magnetism. While we do not fully understand the reason why it is misleading, we clarify them in more detail as "the anisotropic magnetic exchange interactions coming from the spin-orbit coupling".

Reviewer #3: *Eq. S12(previous S13 due to typo) Here, the indexing of the wave vector is in the units of small reciprocal lattice corresponding to large real-space unit cell. On the other hand, indexing in the paramagnetic crystal lattice is used in Eqs S4 and S5. This needs to be properly explained and the relation between these indexing established. Also, the relation of the two expressions shown needs to be described (the second line looks like a second-order perturbation theory?)*

Reviewer #3: *>Eqs. S15 - S19*

Indexing needs to be specified (in which lattice r and q_m, Q_m , are indexed? Do they refer to continuous space, Angstroms and Angstroms inverse? From Eqs. S16, S17, it appears they refer to a specific reciprocal lattice ? explain which one)

Reply: Thank you for pointing out. The reviewer seems to consider that Eq. S12 has ambiguity, but one can take arbitrary k_{μ} at their purpose. In our calculation, we take $k_{\mu}, k_{\nu} = k_x$ or k_y as [1 bar-1,bar-1] and [bar-1,4,bar-5] as we presented in the main text Fig.2(e),

which we also added in the Supplementary.

For some unknown reason, the Reviewer#3 cares/confused with the change of the size of the unit cell in the presence of different magnetic orderings, but the definition of vectors including q_m and Q_m are all commonly determined in unit of the original lattice and the corresponding reciprocal ones. There is no room for ambiguity. To resolve the concerns, we confirm this in the first part of the theoretical section of the Supplementary.

Reviewer #3: >Eq. S14

All quantities (symbols) need to be described

Reply: e_m is the typo, replaced to e^m . We have no other unclarified symbols. Thank you for pointing out.

Reviewer #3: >The U(1) gauge field is constructed using slowly varying sets of $m(r)$.

From the rotation matrix determining the direction of local quantization axes?

Reply: The U(1) vector potential is given in the framework as $A(r) = -\cos\theta(r)\nabla\phi(r)$. (see the definition just below Eq.(5) in the main text.) It is given due to the slowly varying texture of ordered magnetic moments.

List of changes

The changes in the main texts are highlighted.

[Main text]

- Title “ Emergent SU(3) magnons in the thermal Hall effect” → “Emergent SU(3) fluxes on the magnon thermal Hall effect ”
- Abstract: “SU(3) gauge structure” → “SU(3) gauge flux”
“In our description” is added.
- Page 1:
 - “auxiliary fractionalized Majorana fermions”: we add auxiliary.
 - “can bear an emergent U(1) gauge field” → “can be well described as an emergent U(1) gauge field”
 - “to appear as useful in the transport phenomena.” : add “as useful”
 - “to have the anomalous thermal Hall Effect.” → “to describe the anomalous thermal Hall Effect.”
- Page 2: “By performing a large-scale period spin-wave theory, we show that..” → “Our large unit cell spin wave theory calculations show that the heat carriers can be described as the magnons in a complex SU(3) gauge field originating from the three

sublattice structure.”

- Fig.1(a): two layers are clarified, Captions are modified: “The two [111]-planes marked in blue are parallel to each other and include Mn²⁺ ions belonging to different sublattice of the bipartite diamond structure. Inside the plane these ions form a triangular lattice.”
- Discussion section (major revisions):

The 1st paragraphs , expressions are modified to explain the contribution of the lowest few bands to κ_{xy} . (Reviewer#3)

The 2nd paragraph explains the reason for inconsistency between the gauge theory and spin wave in the ferro skyrmions (Reviewer#1).

Page 6-7 (Fig.4) give major revisions about the explanation on the U(1), SU(2) and SU(3) gauges in relevance with several magnetic orderings (Reviewer#1, #3).

We clarified the role of non-commutativity of SU(2) and SU(3) gauges to the gauge invariant fluxes and the reason why it bears the thermal Hall effect.

In Page 7, we clarified more explicitly that 120deg orders (not skyrmions) cannot have SU(3) gauge/flux nor the thermal Hall effect.
- Throughout the text, “energy bands”-> magnon bands”
- Fig.1(d): missing error-bars are added.
- Method: brief explanation on the thermal slope of the setup (Reviewer#2) and the sample dependence of the amplitudes of κ_{xx} and κ_{xy} are added (Reviewer#3) whose details are added in SM.

[Supplementary material]

- Fig.S2 and 2nd and 3rd paragraph of Sec.IA are added (Reviewer #2). “Figure S2 shows...”
- Fig. S4(Previously S3) missing error-bars are added (Reviewer #2).
- Page 2, Sec. IC: Explanation modified. “In MnSc₂S₄, in a paramagnetic phase” (Reviewer #3).
- Page 3, Sec. ID: Explanation modified. “Magnetic skyrmions are only ..” (Reviewer #3).
- Page 3-4, Sec. IE: “Figure S6(a) shows...” (Reviewer #3). “In a high-field phase for B...” explanation on the factors of scaling of κ_{xx} and κ_{xy} are added. (Reviewer #3) to show that it does not deteriorate the reliability the field-dependent part of the data.

The explanation about the domains are also added.
- Page 4, Sec.IIA just after Eq.S3: explanation added (Reviewer#3) about the clarification of notation.
- Page 5, lack of explanation added. Typo corrected.

- Fig. S9: In panel (c) the data previously presented in Fig.4 in the main text is transferred due to the lack of space in the main text.

REVIEWER COMMENTS

Reviewer #1 (Remarks to the Author):

The authors have addressed my comments well. I only have one minor comment now. Has the SU(2) magnon thermal Hall effect been experimentally observed? I know that the authors have proposed some candidates in theory. If the SU(2) magnon thermal Hall effect has not been achieved, the authors may want to mention that it is interesting to observe it in experiments somewhere.

Reviewer #2 (Remarks to the Author):

The revised version solved my concerns. I could recommend for the publication.

Reviewer #3 (Remarks to the Author):

While authors have addressed some of the minor comments listed in my previous report in the revised manuscript, many issues remain. I do not find authors' rebuttal of my criticism convincing and therefore cannot recommend the manuscript for publication.

1. My main concern was and still is with the vastly different magnitude of the transverse heat conductance in two samples. Authors' response to my concern that this "can scarcely occur as an extrinsic effect" is unconvincing. It also appears that authors misunderstand the concept of an extrinsic effect in the measured transport coefficients. Extrinsic are effects which are present in the sample/measurements but are not included in theoretical understanding/description of the measured quantity. Such are effects of scattering by impurities if they are not included in the description, and if these effects lead to a variation of the measured quantity by a factor 3-10, they need to be included in theoretical description. Otherwise, quantitative theoretical description appears not possible. As an example, authors might consider electrical conductivity of a metal, another transport coefficient measured in a conceptually similar "out-of-equilibrium" setup. Low-temperature conductivity is determined by scattering of the electrons on impurities and can differ vastly, by orders of magnitude, in samples of different purity. However, this effect is well understood and can be accounted for in theory. A similar account is needed in the case of thermal transport presented in the manuscript, which is missing. Addressing the specifics of authors' response to this concern, authors argue that the field-dependent fraction of the measured transverse thermal conductance should be scaled down for sample #2 by a factor ~ 3 that scales down the diagonal phonon thermal conductivity of sample 2 relative to sample 1. I do not "buy" this argument: why would the transverse thermal conductance of magnons need to be scaled by thermal conductivity of phonons? These seem to be unrelated?

>In various magnetic insulators, the sample with smaller amplitude of k_{xx} is found to have smaller fraction of the magnetic components.

What is "magnetic component" discussed here? If this refers to the component of thermal conductivity assigned to magnons, then it would appear to have been already adequately taken into account through scaling by a factor 3.2 which corresponds to the relative amplitude of the field-dependent k_{xx} of the two samples. Then, it is quite unclear what is the reason for the additional factor 3 between the transverse thermal conductivities of the two samples. Why would field-dependent k_{xx} and k_{xy} , both carried by magnons, scale differently?

Authors argue that a suppression factor ~ 3 which they ascribe to the increased phonon scattering which does not influence the longitudinal magnon heat conductance should nevertheless be applied to the transverse part of magnon heat conductance. I do not find any convincing arguments for doing so neither in the text of manuscript and SI, nor in authors' reply to my previous review. In my view, both longitudinal and transverse magnon heat conductance of different samples should scale by the same scale factor, which probably depends on the sample quality. This seem to be in agreement with the

authors' own assessment that,

> broadening effect of magnon bands due to the scattering effect on phonons influences both κ_{xx} and κ_{xy} in our case ...

2. My second concern was about misrepresenting the proposed theoretical description of the measured transverse heat conductance as the central finding of the manuscript, rather than the measurement itself. Instead of applying the required revisions in the manuscript, authors decided to "argue away" this issue.

In their response authors write that "Magnons in insulating magnets are simple bosonic excitations" is a "definition of a magnon". This seems to be one of many misunderstandings. By definition, magnons are quasiparticles carrying quantum numbers of energy, momentum, and angular momentum (spin). As I explained in my previous review, while magnons in insulating magnets are most often described as "simple bosonic excitations", in some cases they can be alternatively described as fermions, or anions coupled to gauge field (see E. Fradkin, Phys. Rev. Lett. 63, 322 (1989) for two-dimensional spin-1/2 case; see also Affleck's (bosonic) vs Tsvetlik's (fermionic) description of spin-1 chains, etc). In some cases, bosonic description of magnons might be more useful, in others - fermionic, or anyonic, with corresponding gauge fields. Instead of revising their text accordingly, authors chose to engage in lengthy arguments which seem to deny these well-established descriptions. The important corollary of this point, which I also explained in the previous report and which authors chose to rebut rather than take into account, is that gauge fields that might appear in a theoretical description do not necessarily represent any kind of physical reality - just a computational convenience.

Below, I reiterate several other comments that were included in my previous report but not addressed in the revised manuscript.

Abstract: > the magnon excitations can contribute to anomalous transports by feeling the U(1) and SU(2) fluxes arising from the features of ordered moments or interactions.

Should be: the magnon contribution to anomalous transports can be described in terms of U(1) and SU(2) fluxes present in the ordered magnetic structure

>Here, we report an emergent higher rank SU(3) flux in the magnon transport based on the thermal conductivity measurements of MnSc₂S₄ in an applied field up to 14 T

Should be: Here, we report thermal conductivity measurements of MnSc₂S₄ in an applied field up to 14 T, which we describe in terms of an emergent higher rank SU(3) flux in the ground state controlling the magnon transport

Authors: >which is confirmed by the large-scale spin wave theory

Should be: which is consistent with classical spin-wave theory

>We also add a tiny constant to ϵ_r by hands to support the positive definite spectrum of the magnon bands

That a spectrum obtained through diagonalization of a spin-wave Hamiltonian based on the assumed magnetic structure (which is accounted for via spin rotations to the local frames prior to Holstein-Primakoff transformation) indicates that the assumed ground state is incorrect. For example, the optimal magnetic structure in magnetic field becomes incommensurate and not adequately described by using even such a large magnetic unit cell as authors do. This is a serious handicap and cannot be simply "cured" by adding a constant term to shift the spectrum out of negative domain as authors seem to be doing here (as well as in Ref. 8). Is this shift same for all magnetic fields? - The true spin ground state of the "original Hamiltonian" changes as a function of magnetic field and so its possible deviation from the multi-sublattice commensurate structure assumed by the authors. The seriousness of this problem and the accuracy of the proposed "ad hoc" approximation need to be discussed.

In response to my request to define the indexing of the wave vectors used in the manuscript, authors included the following statement,

>The coordinates of the vectors are all commonly defined based on the original lattice (before the magnetic ordering in a unit of cell spacings), and so as the reciprocal ones.

This is an awkward, unclear, and quite unusual way to describe the indexing used - rather, it states some sort of authors belief concerning how "vectors are all commonly defined", which is not a very useful information for the reader. Instead, authors should include a short sentence stating:

>All vectors are indexed in the unit cell of the paramagnetic crystal lattice of MnSc₂S₄ (space group 227, Fd-3m, lattice parameter $a=10.606$) and wave vectors are indexed in units of the corresponding reciprocal lattice."

Such a statement is short, clear, useful for the reader, and is properly fitting in the context of a scientific manuscript.

Small stylistic comments:

Authors: >Interestingly, the phases above TN may not be a simple paramagnet but a classical spiral-spin liquid

Should be: Interestingly, the phase above TN is a correlated paramagnet, which can also be described as a classical spiral-spin liquid

While authors write in their response that they have revised an awkward and unclear "large-scale spin-wave theory" terminology, I find that it still proliferates in the manuscript, eg in the paragraph with Eq. 1 (and elsewhere).

Reply to Reviewer#1

Reviewer#1: The authors have addressed my comments well. I only have one minor comment now. Has the SU(2) magnon thermal Hall effect been experimentally observed? I know that the authors have proposed some candidates in theory. If the SU(2) magnon thermal Hall effect has not been achieved, the authors may want to mention that it is interesting to observe it in experiments somewhere.

Reply: We appreciate Reviewer#1 for recognizing our theoretical work. Thanks to the comment, we started to think about the possibility of already published experimental works to fit to the SU(2) magnon thermal Hall effect, and arrived at Na₂Co₂TeO₆, in Phys. Rev. Research 4, L042035 (2022). When the thermal Hall experiment was performed, it was discussed differently, but taking a closer look, its antiferromagnetic phase with two sublattice, and part of thermal Hall effect might be interpreted as generating an SU(2) flux. Because the model Hamiltonian is different from our previous theory on square lattice antiferromagnet, a more careful examination is needed, and we briefly point out the possibility in the main text. Thank you for the valuable suggestion.

Reply to Reviewer#3

We thank Reviewer#3 for your efforts of reading our long letter and for expanding the comments in detail about your concerns.

[We first clarify the most important point 1]

Reviewer#3 argues that the main concern is not resolved, which regards the experimental reliability related to the largely different magnitude of the thermal transport coefficient between the two samples.

We read your letter very carefully, and found that we are having an unfortunate miscommunication, but in reality, the above-mentioned issue does not exist.

We admit that our previous explanation was indeed circuitous and caused the problem. Because we deeply regret it, we first try to clarify the issue directly in a data-based manner.

Reviewer# 3 mentions the core of his/her question as,

“Then, it is quite unclear what is the reason for the additional factor 3 between the transverse thermal conductivities of the two samples. Why would field-dependent k_{xx} and k_{xy} , both carried by magnons, scale differently?”

To this question, our experimental fact is:

- $\Delta\kappa_{xx}(B) = \kappa_{xx}(B) - \kappa_{xx}(0)$, a field-dependent part of κ_{xx} has [sample#1/sample#2] = 10,
- $\kappa_{xy}(B)$, (here, $\kappa_{xy}(0) = 0$), has, [sample#1/sample#2] = 10.

The scaling factors of the *raw amplitude* of two quantities carried by magnons have the same scaling factor, and we plot here below to show it directly. All the data are the same as

those of previous manuscript in Fig.S6, where we have plotted differently as $\Delta\kappa_{xx}(B)/\kappa_{xx}(0)$. Therefore, to conclude, Reviewer#3's concern does not apply to our data, and this concern can be immediately resolved from the above figure. Our previous explanation was two-fold: it was not the good way, and confused Reviewer#3, which we regret a lot.

To be sure, our previous figure showed $\Delta\kappa_{xx}(B)/\kappa_{xx}(0)$, not the absolute value of $\Delta\kappa_{xx}(B)$ and we wrote, (see SM and Fig. S6 (a)-(c))

- (a) $\kappa_{xx}(0)$ is [sample# 1/ sample#2] = 3
- (b) $\Delta\kappa_{xx}(B)/\kappa_{xx}(0)$ is [sample# 1/ sample#2] = 3.2
- (c) $\kappa_{xy}(B)$ is [sample# 1/ sample#2] = 10

Based on (a)-(c) ($\Delta\kappa_{xx}(B)$ of #1) / ($\Delta\kappa_{xx}(B)$ of #2) = 3.2 x 3 = 9.6 ~ 10, namely, we concluded that $\Delta\kappa_{xx}(B)$ and $\kappa_{xy}(B)$ have both [sample# 1/ sample#2] ~ 10.

To explain the reason why the absolute value of $\kappa_{xx}(0)$ (for zero field) in (a) generates factor 3, we explained that phonon mean free path or the phonon scattering differs between sample #1 and sample #2, but it is nothing to do with the field-dependent part itself. We have explained about the factor 3 in (a) because it is the denominator of the plotted value $\Delta\kappa_{xx}(B)/\kappa_{xx}(0)$.

Therefore, for the rest of the comments by Reviewer#3, we fully agree, if it were telling us that the scaling factors sample#1/sample#2 of $\kappa_{xx}(0)$ and $\kappa_{xy}(B)$ have no reason to be related. Indeed, they are not quantitatively related. We find it as a substantial effort to explain us how he/she thinks and thank Reviewer#3 for it.

In SM Fig.S6, we added Fig.S6(d) the absolute value $\Delta\kappa_{xx}(B)$ with scale factor 10 for sample #2, which are the same data as Fig.S6(b) $\Delta\kappa_{xx}(B)/\kappa_{xx}(0)$ plotted with scale factor 3.2. We clarified the related explanation.

Minor replies to the comments: With the above clarification, we basically believe that the following issues are resolved. (To confirm, we show point-by-point)

Reviewer#3: Addressing the specifics of authors' response to this concern, authors argue that the field-dependent fraction of the measured transverse thermal conductance should be scaled down for sample #2 by a factor ~3 that scales down the diagonal phonon thermal conductivity of sample 2 relative to sample 1. I do not "buy" this argument: why would the transverse thermal conductance of magnons need to be scaled by thermal conductivity of phonons? These seem to be unrelated?

Reply: The scaling factor ~3 is for phonons $\kappa_{xx}(0)$, in the above explanation (a). It is not related to magnons nor to field-dependent part in (c) and (d).

Reviewer#3: (author) "In various magnetic insulators, the sample with smaller amplitude of κ_{xx} is found to have smaller fraction of the magnetic components."

What is "magnetic component" discussed here?

If this refers to the component of thermal conductivity assigned to magnons, then it would appear to have been already adequately taken into account through scaling by a factor

3.2 which corresponds to the relative amplitude of the field-dependent κ_{xx} of the two samples.

Then, it is quite unclear what is the reason for the additional factor 3 between the transverse thermal conductivities of the two samples. Why would field-dependent κ_{xx} and κ_{xy} , both carried by magnons, scale differently?

Reply: As mentioned in the first part, field-dependent part refers to $\Delta\kappa_{xx}(B)$ (absolute value) and κ_{xy} . They have the same scaling factors of ~ 10 , not 3.2. There is no difference.

Reviewer#3: Authors argue that a suppression factor ~ 3 which they ascribe to the increased phonon scattering which does not influence the longitudinal magnon heat conductance should nevertheless be applied to the transverse part of magnon heat conductance.

I do not find any convincing arguments for doing so neither in the text of manuscript and SI, nor in authors' reply to my previous review. In my view, both longitudinal and transverse magnon heat conductance of different samples should scale by the same scale factor, which probably depends on the sample quality. This seem to be in agreement with the authors' own assessment that,

> broadening effect of magnon bands due to the scattering effect on phonons influences both κ_{xx} and κ_{xy} in our case ...

Reply: We again confirm that "both longitudinal and transverse magnon heat conductance of different samples should scale by the same scale factor": this holds in our data.

We refer to the broadening effect here not to explain about the scaling factors. We were simply referring to the overall tendency and the relation with the sample quality. We mention that in previous studies, samples with smaller $\kappa_{xx}(0)$ have smaller $\kappa_{xy}(B)$ and $\Delta\kappa_{xx}(B)$ as well. We have no intention to add any quantitative explanation on it.

We clarified it in the updated SI.

Reviewer#3: (author's) " $(\kappa_{xx}(B) - \kappa_{xx}(0))/\kappa_{xx}(0)$, presented in Fig.2 can scarcely occur as an extrinsic effect." is unconvincing.

It also appears that authors misunderstand the concept of an extrinsic effect in the measured transport coefficients. Extrinsic are effects which are present in the sample/measurements but are not included in theoretical understanding/description of the measured quantity.

Reply: In the following sentence, we use "extrinsic" and "intrinsic" in the way Reviewer#3 states. The component $\Delta\kappa_{xx}(B)$, is induced by field, and is natural to ascribe to magnon as an intrinsic effect. However, $\kappa_{xx}(0)$ suffers the extrinsic effect. Therefore, $\Delta\kappa_{xx}(B)/\kappa_{xx}(0)$ includes both, and our expression was confusing. We clarified the precise definition of these quantities in the updated SI.

There is a usage of specific terminology "extrinsic" possibly as a jargon in a field of anomalous transport measurements: "intrinsic mechanism" (coming from Berry phase effects) and "extrinsic mechanism" (by either skew or side-jump scatterings) are used in

many related articles.

We agree that in a wider field, as Reviewer#3 mentions, extrinsic scattering effects can be incorporated in phenomenological parameters like quasi particle lifetime and mean free path, commonly for electrons, phonons, and magnons.

[Concerns on the theoretical description, point 2]

We reply to the comments about (1st point) the terminology, (2nd point) about some numerical details given in SI.

(1st point) Reviewer#3: My second concern was about misrepresenting the proposed theoretical description of the measured transverse heat conductance as the central finding of the manuscript, rather than the measurement itself. Instead of applying the required revisions in the manuscript, authors decided to "argue away" this issue.

In their response authors write that "Magnons in insulating magnets are simple bosonic excitations" is a "definition of a magnon". This seem to be one of many misunderstandings. By definition, magnons are quasiparticles carrying quantum numbers of energy, momentum, and angular momentum (spin). As I explained in my previous review, while magnons in insulating magnets are most often described as "simple bosonic excitations", in some cases they can be alternatively described as fermions, or anions coupled to gauge field (see E. Fradkin, Phys. Rev. Lett. 63, 322 (1989) for two-dimensional spin-1/2 case; see also Affleck's (bosonic) vs Tsvetik's (fermionic) description of spin-1 chains, etc). In some cases, bosonic description of magnons might be more useful, in others - fermionic, or anyonic, with corresponding gauge fields. Instead of revising their text accordingly, authors chose to engage in lengthy arguments which seem to deny these well-established descriptions.

The important corollary of this point, which I also explained in the previous report and which authors chose to rebut rather than take into account, is that gauge fields that might appear in a theoretical description do not necessarily represent any kind of physical reality - just a computational convenience.

Reply: Thank you for your comment. We first would like you to understand that we are not against your comment, particularly in that "*gauge fields that might appear in a theoretical description do not necessarily represent any kind of physical reality*".

We are not trying to "*argue away*", but want to present in our paper that SU(3) gauge flux gives a core picture (not necessarily a reality) to interpret to the most simplest, why the theoretical calculation (numerical) and experimental measurements conform to each other.

We may have given the impression that we are "arguing away" in the way we wanted to address that we cannot agree if Reviewer#3 were asking us to remove all these discussions from the manuscript. However, after reading the second report, we now recognize that it was our precaution.

Related to the above comments, Reviewer#3: Below, I reiterate several other comments that were included in my previous report but not addressed in the revised manuscript.

[Three sentences in the abstracts,]

[In response to the request to define the indexing of the wave vectors used in the manuscript, ... given a proposal.]

[Small stylistic comments: ...about a classical spiral-spin liquid and large scale spin wave theory]

Reply: Thank you very much for telling us the proper way of addressing the information briefly and accurately, which we were not recognizing.

We revised all these parts, the three sentences in the abstract, the term “spin liquid” and “large scale spin wave”, and the description of the wave vectors in the SI,

For details, please find the List of modifications attached in the last part of the reply.

(2nd point) *Reviewer#3:*

>We also add a tiny constant to ϵr by hands to support the positive definite spectrum of the magnon bands. That a spectrum obtained through diagonalization of a spin-wave Hamiltonian based on the assumed magnetic structure (which is accounted for via spin rotations to the local frames prior to Holstein-Primakoff transformation) indicates that the assumed ground state is incorrect. For example, the optimal magnetic structure in magnetic field becomes incommensurate and not adequately described by using even such a large magnetic unit cell as authors do. This is a serious handicap and cannot be simply "cured" by adding a constant term to shift the spectrum out of negative domain as authors seem to be doing here (as well as in Ref. 8). Is this shift same for all magnetic fields? - The true spin ground state of the "original Hamiltonian" changes as a function of magnetic field and so its possible deviation from the multi-sublattice commensurate structure assumed by the authors. The seriousness of this problem and the accuracy of the proposed "ad hoc" approximation need to be discussed.

Reply: We may first resolve the worries of Reviewer#3.

In a magnetic field the *periodicity* of the present magnetic structure given in Eqs. (S4) and (S5) is not destroyed. Namely, there is no transformation to different incommensurate structure. This fact is also confirmed in the previous Monte Carlo study for wide parameter range in Ref. [24].

The tiny constant added to ϵr in Eq.(S9) is not to prevent the instability to different periodicity but to exclude the higher order harmonics, a small secondary effect that appear on top of the strongest wave number. If we include this secondary effect seriously, some small modification of the magnetic moment inside the unit cell occurs, which may depend on the field. However, this effect is quantitatively small and is safely neglected, and its error should be within the difference between the Eqs. (S4) and (S5) and the Monte Carlo simulation.

Adding very small constant and excluding such small change can safely and largely simplify the treatment, while keeping the influence on the magnon bands to be subtle. The added tiny constant does not depend on the field because it is just to exclude the secondary one.

Because our treatment is anyway a linear spin wave theory that starts from the classical spin configuration, the "ad hoc" approximation is present, which audiences can judge. Our small approximation about the magnetic structure does not change much the degree of *ad hoc* ness. According to the request, we added the explanation in the Supplementary material.

Thank you for carefully reading the content.

[List of revisions]

Main text

- Abstract

"the magnon contribution to anomalous transports can be described in terms of $U(1)$ and $SU(2)$ fluxes present in the ordered magnetic structure."

"Here, we report thermal transport measurements of $MnSc_{2}S_{4}$ in an applied field up to $14T$, which we describe in terms of an emergent higher rank $SU(3)$ flux, controlling the magnon transport."

"which is consistent with linear spin-wave theory."

Reply: We agree with the suggested revision.

Since $SU(3)$ flux does not appear in the "ground state" (only in the excitation), we deleted it, which does not change the main suggestion.

- *"which is confirmed by the large-scale spin wave theory"*

→ *"which is consistent with linear spin-wave theory"*

Reply: We avoid "large scale" as proposed but prefer to use a more formal "linear spin wave theory" than "classical spin wave" suggested.

- *"Interestingly, the phases above T_N may not be a simple paramagnet but a classical spiral-spin liquid"*

→ *"Interestingly, the phase above T_N is a correlated paramagnet, which can be described as a classical spiral-spin liquid,"*

Reply: We followed the advice.

- "Large-scale spin wave theory"

→ *"Linear spin wave theory in a large unit cell"*

Reply: We eliminated "large-scale" for all parts in the text which we have overlooked. For the title of the section we added "in a large unit cell".

- Page 7: experimental work on $\text{Na}_2\text{Co}_2\text{TeO}_6$ is newly cited as Ref.[50],.
(related to Reviewer#1's request.)
- Method section:
"The amplitude of κ_{xx} and κ_{xy} differ between sample #1 and #2 consistently by a factor of 9-10, "
 → *"The amplitude of $\Delta\kappa_{xx}(B) = \kappa_{xx}(B) - \kappa_{xx}(0)$ and $\kappa_{xy}(B)$ differ between sample #1 and #2 consistently by a factor of 9-10, "*

Comment: We apologize for this misleading and careless expression.

Supplementary material

- Experimental: regarding the sample dependence.
Fig.S6 (d) added, related explanations in page 3, right column is revised.
- Theoretical: sentence added according to the suggestions about the indexing of the wave vectors in page 4.
"Throughout this work, the coordinates of the vectors are..."
- Theoretical: page 5, explanation added about the small epsilon, according to the request.
"For practical purpose, we add a tiny constant to ϵ_r by..."

REVIEWERS' COMMENTS

Reviewer #3 (Remarks to the Author):

1. I am satisfied with authors' response to my concern related to different scaling factors in their measurements on the two samples and the corresponding revisions in the manuscript. The presentation in the original manuscript and in the authors' first response to my comments were really confusing. Now the issue is cleared in the revised Supplementary and the amended Figure S6. This point is therefore resolved.

2. While authors have positively responded to my comment that manuscript requires substantial presentational changes, they have not done enough to implement the corresponding revisions in the manuscript. The point of the requested revisions is to properly focus the presentation of the manuscript on the reported experimental results rather than on their theoretical interpretations, which are of secondary importance and, as authors properly acknowledge, have high degree of "ad hoc"-ness. Frustratingly, authors have mainly only included changes which I wrote down explicitly as an example of how to revise. Are authors expecting the reviewer to revise the entire manuscript for them rather than do it themselves following the provided guidelines? This is an important point, and the manuscript should be revised to address this point. Below, I list several more examples highlighting the essence of the required revisions, but this list is not complete. In addition to implementing these exemplary revisions, authors should go over the text and revise it in accordance with the guidelines these revisions provide. H

- In order to adequately summarize the reported results, the title of the manuscript should be changed to something like

"Magnon thermal Hall effect and emergent SU(3) flux on the antiferromagnetic skyrmion lattice"

or

"Magnon thermal Hall effect via emergent SU(3) flux on the antiferromagnetic skyrmion lattice"

- Abstract.

"Complexity of quantum phases of matter is often understood by the underlying gauge structures" 

"Complexity of quantum phases of matter is often understood theoretically by using gauge structures", or something like that.

-Text:

"... Hall conductivity reported in α -RuCl₃ is argued as being carried by these auxiliary fractionalized ..."  "... Hall conductivity reported in α -RuCl₃ can be described as being carried by these auxiliary fractionalized ..."

and similarly, elsewhere, with the goal of properly delineate the physical reality, such as measurable Hall conductivity, and convenient theoretical constructs, such as gauge fields, quasiparticles, etc.

3. My final major concern was about theoretical description where spin-wave theory requires an ad hoc shift of the obtained spectra, indicating an incorrect ground state. In their response, authors apparently confirm that this is indeed the case and that the correct ground state would include harmonics of the main magnetic wave vector. Firstly, methods of accounting for these type of harmonic corrections to the classical spin ground state and the ensuing spin-wave spectra are well-established [see eg Phys. Rev. B 53, 3428 (1996), or book "Spin-wave Theory and Its Applications to Neutron Scattering and THz Spectroscopy" by Fishman, et al (2018)]. These previous results show how harmonic components scale with magnetic field an anisotropy and allow to quantitatively evaluate the degree of validity of their neglect. Importantly, however, the presence of harmonics in the ground state structure causes gaps in the spin-wave dispersion at the corresponding harmonic positions [see eg Phys. Rev. B 59, 11432 (1999)] in the description based on paramagnetic unit cell. In the description based on magnetic supercell, these terms cause repulsion and ensuing gaps of the otherwise degenerate spin-wave branches at the Brillouin zone center/boundary. Therefore, even if

small, these terms can cause qualitative changes to the topological characteristics of magnon spectra. While neglect of these harmonic effects might still be justified, it needs to be thoroughly substantiated.

To summarize, I think that authors have addressed large part of comments listed in my previous reports, and that this has improved the quality of the manuscript notably. However, several important issues remain. Authors need to address these issues before the manuscript can be accepted for publication in Nature Communicaitons.

Reply to Reviewer#3's comment:

----- List of changes

For comment 2:

- We changed the title to :

"Magnon thermal Hall effect via emergent SU(3) flux on the antiferromagnetic skyrmion lattice"

- We changed the text:

"... Hall conductivity reported in α -RuCl₃ is argued as being carried by these auxiliary fractionalized ..."

 " Hall conductivity reported in α -RuCl₃ is argued to fit the picture of being carried by these auxiliary fractionalized Majorana fermions"

"In chiral magnets such as MnSi... the conduction electrons feel an emergent gauge field as they travel through the spatially varying spin texture and contribute to the topological Hall effect. "

"the conduction electrons feel an emergent gauge field as they travel through the spatially varying spin texture, which gives a good interpretation of the topological Hall effect".

"Magnons in insulating magnets are simple bosonic excitations but can also carry a U(1) gauge"

 "Magnons in insulating magnets are dealt as simple bosonic excitations but can also carry a U(1) gauge"

For comment 3:

In supplementary page 5 second column, we added:

"However, its order is of that tiny constant $O(0.01)$, and even if the higher energy part of the magnon band may slightly change at this order, the redistribution of Berry curvature and its influence on the thermal Hall coefficients is minor because of the Bose statistics. "

Reply to the other two comments:

2. In addition to the correction suggested, we have changed two extra parts on the first page to cope most with the inquiry.

As mentioned, we have already revised many parts pointed out by Reviewer#3, and we do not think that there is any more remaining issue.

In the present manuscript, we find no room for the audiences to misinterpret or confuse the experimental measurement as "the physical reality" and the theory as providing a convenient picture and understanding.

Spin waves, concepts of gauges, and flux are used in many articles as theoretical tools, and their bound is already well established and known to wide audiences.

We have tried to avoid repeated usage of "described by" or "expressed by" which are naturally felt to be redundant for most of the audiences and make the sentence difficult to read.

In particular, they are useless in the purely theoretical part, where the gauges and pictures are facts within the formulation or approximation.

While we understand Reviewer#3's concern, the choice of expressions used in the paper, if it is not scientifically wrong beyond the level of personal opinion, belongs to the important part of the authorship.

We had three Reviewers, and the opinions about the expressions were coming only from Reviewer#3, indicating that there are variants in the audiences and the manuscript falls within that range.

We are grateful with the criticisms raised by Reviewer#3 which helped to improve our manuscripts, which we are satisfied, and are welcome to accept the judgment of future audiences about the way the manuscript is written.

3. About the comment: "My final major concern was about theoretical description where spin-wave theory requires an ad hoc shift of the obtained spectra, indicating an incorrect ground state. "

As we have clarified in the previous reply, the small constant added to e_r of $O(0.01)$ does not change the result. The magnetic moment within the unit cell m_r given in Eq.S4, S5 and the Monte Carlo simulations are consistent. The error as we have mentioned is of the order of the difference between Eqs. S4, S5 and the Monte Carlo ones.

In fact, the order of the tiny constant added is $O(0.01)$, which if influences the gap, would not discriminate in the magnon band structure in the present scale of calculation and approximation.

Regarding this point, Reviewer#3 mentions that "these terms cause repulsion and ensuing gaps of the otherwise degenerate spin-wave branches at the Brillouin zone center/boundary. Therefore, even if small, these terms can cause qualitative changes to the topological characteristics of magnon spectra."

This is a misunderstanding.

We can say that a tiny, almost discernable gap may open at a very high energy level, but usually, the summation of Berry curvatures over upper and lower bands is not going to change with it. In the thermal Hall conductivity, the contribution is indeed almost a simple summation of Berry curvature over the upper and lower bands at the crossing or the gap because the Bose distribution function does not change within that high energy range for the target temperature. The major part of the phenomena comes from the magnetic structure in Eqs. S4, S5 (an SU(3) gauge flux is not influenced by that small fluctuation of $m(r)$). This point is different from electronic systems where the gap opening and sign change of the Berry curvature near the Fermi level qualitatively change the Hall conductivity.